

# Evidence for renoxification in the tropical marine boundary layer
**Chris Reed[1†], Mathew J. Evans[1,2], Leigh R. Crilley[3], William J. Bloss[3], Tomás Sherwen[1],**
**Katie A. Read[1,2], James D. Lee [1,2] and Lucy J. Carpenter[1]**
[1] Wolfson Atmospheric Chemistry Laboratories (WACL), Department of Chemistry,
University of York, Heslington, York, YO10 5DD, United Kingdom
[2] National Centre for Atmospheric Science (NCAS), University of York, Heslington, York,
YO10 5DD, United Kingdom
[3] School of Geography, Earth and Environmental Sciences, University of Birmingham,
Edgbaston, Birmingham, B15 2TT, United Kingdom
Corresponding author: Chris Reed (chris.paul.reed@gmail.com)
†Now at Facility for Airborne Atmospheric Measurements (FAAM), Building 146, Cranfield
University, Cranfield, MK43 0AL, United Kingdom
**Abstract**
We present two years of $NO_x$ observations from the Cape Verde Atmospheric Observatory
located in the tropical Atlantic boundary layer. We find $NO_x$ mixing ratios peak around solar
noon (at 20-30 pptV depending on season), which is counter to box model simulations that show
a midday minimum due to OH conversion of $NO_2$ to $HNO_3$. Production of $NO_x$ via
decomposition of organic nitrogen species and the photolysis of $HNO_3$ appear insufficient to
provide the observed noon-time maximum. A rapid photolysis of nitrate aerosol to produce
HONO and $NO_2$, however, is able to simulate the observed diurnal cycle. This would make it the
dominant source of $NO_x$ at this remote marine boundary layer site overturning the previous
paradigm of transport of organic nitrogen species such as PAN being the dominant source. We
show that observed mixing ratios (Nov-Dec 2015) of HONO at Cape Verde (~2.5 pptV peak at
solar noon) are consistent with this route for $NO_x$ production. Reactions between the nitrate
radical and halogen hydroxides which have been postulated in the literature appear to improve
the box model simulation. This rapid conversion of aerosol phase nitrate to $NO_x$ changes our
perspective of the $NO_x$ cycling chemistry in the tropical marine boundary layer, suggesting a
more chemically complex environment than previously thought.





**1      Introduction**
The chemical environment in the remote marine boundary layer (MBL) is characterized by very
low concentrations of nitrogen oxides ($NO_x = NO + NO_2$), high concentrations of water vapour
and the presence of inorganic halogen compounds, resulting in net daytime ozone ($O_3$)
destruction (Dickerson et al., 1999; Read et al., 2008; Sherwen et al., 2016; Vogt et al., 1999).
This MBL loss of ozone plays an important role in determining the global budget of ozone and
the overall oxidizing capacity of the region. Understanding the concentrations of $NO_x$ in these
environments is thus important for determining the global ozone budget, alongside wider
atmospheric chemical impacts.
$NO_x$ in the remote MBL has been attributed to a) long range transport and decomposition of
species such as PANs, organic nitrates, or $HNO_3$ (Moxim et al., 1996) b) shipping emissions
(Beirle et al., 2004) c) a direct ocean source (Neu et al., 2008) and d) its direct atmospheric
transport (Moxim et al., 1996). However, more recently the possibility of 'renoxification' by
particulate nitrate photolysis has garnered attention (Baergen and Donaldson, 2013; Cohan et al.,
2008; Handley et al., 2007; Ndour et al., 2009; Scharko et al., 2014; Ye et al., 2016a, 2016b;
Zhou et al., 2003).
The oxidation of $NO_2$ to $HNO_3$ by OH is through the predominant sink for NOx in the remote-
MBL. $NO_x$ can also be converted into aerosol phase nitrate via the hydrolysis of $N_2O_5$ (R2)
(Evans and Jacob, 2005) but this is slow in these low $NO_x$ environments. $NO_x$ can be returned
through $HNO_3$ photolysis (R3) or reaction with OH (R4) but in general these processes are slow
and so $HNO_3$ can deposit to the surface, be washed out by rain, or taken up by aerosol (R5).
$NO_2 + OH + M \rightarrow HNO_3 + M$ (R1)
$N_2O_5 + H_2O_{(aer)} \rightarrow 2HNO_{3(aer)}$ (R2)
$HNO_3 + hv \rightarrow OH + NO_2$ (R3)
$HNO_3 + OH \rightarrow NO_3 + H_2O$ (R4)





$HNO_{3(g)} + aerosol \rightarrow HNO_{3(aer)}$ (R5)
More recently the production and subsequent hydrolysis of halogen nitrates ($IONO_2$, $BrONO_2$,
$ClONO_2$) have been suggested to be a potentially important sink for $NO_x$ in the marine boundary
layer (Keene et al., 2007, 2009; Lawler et al., 2009; Pszenny et al., 2004; Sander et al., 1999)
In this paper we investigate the budget of $NO_x$ in the remote MBL using observations of $NO_x$
and HONO collected at the Cape Verde Atmospheric Observatory during 2014 and 2015. We
use a 0-D model of $NO_x$, $HO_x$, halogen, and VOC chemistry to interpret these observations and
investigate the role that different $NO_x$ source and sink terms play.

## 2    Methodology

The Cape Verde Atmospheric Observatory (CVO), a global WMO Global Atmospheric Watch
(GAW) station, is located in the tropical North Atlantic (16.864, -24.868) on the island of São
Vincente and is exposed to air travelling from the North East in the trade winds (Carpenter et al.,
2010). In general, the air reaching the station has travelled many days over the ocean since
exposure to anthropogenic emissions, thus the station is considered representative of the remote
marine boundary layer (Read et al., 2008). A large range of compounds are measured at the
CVO (Carpenter et al., 2010), but we focus here on the NO and $NO_2$ continuous measurements,
alongside HONO measurements that were made for a short period in Nov/Dec 2015.

### 2.1    NO and $NO_2$

NO and $NO_2$ are measured by NO chemiluminescence (Drummond et al., 1985) coupled to
photolytic $NO_2$ conversion by selective photolysis at 385-395 nm as described by (Lee et al.,
2009; Pollack et al., 2011; Reed et al., 2016a, 2016b; Ryerson et al., 2000). A single
photomultiplier detector switches between 1 minute of chemiluminescent zero, 2 minutes of NO,
and 2 minutes of $NO_x$ measurement. Calibration for NO sensitivity and $NO_2$ converter efficiency
occurs every 71 hours in ambient air; in this way correction for humidity affecting sensitivity,
and $O_3$ affecting $NO_2$ conversion efficiency are unnecessary. The humidity of the sample gas
reduced by a Nafion dryer (PD-50T-12-MKR, Permapure), fed by a constant sheath flow of zero
air (PAG 003, Eco Physics AG) which is also filtered through a Sofnofil (Molecular Products)



and activated carbon (Sigma Aldrich) trap. This zero air is also used to determine the $NO_2$
artifact signal which can arise when $NO_x$ free air is illuminated at UV wavelengths due to
photolysis of $HNO_3$ etc., adsorbed on the walls of the photolysis cell (Nakamura et al., 2003;
Pollack et al., 2011; Ryerson et al., 2000). NO artifact correction is made by assuming it is
equivalent to a stable night-time NO value in remote regions (Lee et al., 2009), away from any
emission source, where NO should be zero in the presence of $O_3$. Reed et al., (2016b) showed
that thermal interferences in $NO_2$ using this technique may cause a bias in cold or temperate
remote regions, but that in warm regions, such as Cape Verde, the effect is negligible. Photolytic
interferences such as $BrONO_2$ and HONO, and inlet effects may also alter the retrieved NO or
$NO_2$ (Reed et al., 2016a, 2016b). These effects are considered to be sufficiently small that the
concentrations of NO and $NO_2$ can be determined within an accuracy of 5% and 5.9%
respectively (Reed et al., 2016a, 2016b) at the (very low) levels present at CVO. The instrument
having a zero count rate of ~ 1700 Hz with 1 $\sigma$ standard deviation of that signal being ~ 50 Hz
this gives a precision of 7.2 pptV for 1 second data with typical sensitivity over the measurement
period of 6.9 cps/pptV. The resultant limits of detection for NO and $NO_2$ being 0.3 and 0.35
pptV when averaged over an hour.
**2.2    HONO**
Between 24th November and 3rd December 2015 a Long Path Absorption Photometer (LOPAP)
(Heland et al., 2001) was employed at CVO to provide an *in-situ* measurement of nitrous acid.
The LOPAP has its own thermostated inlet system with reactive HONO stripping to minimise
losses so did not sample from the main lab manifold. The LOPAP inlet was installed on the roof
of a container lab ~ 2.5 m above ground level, unobstructed from the prevailing wind.
Calibration and operation of the LOPAP was carried out in line with the standard procedures
described by Kleffmann and Wiesen, (2008). Further details of the HONO measurement
approach can be found in (Crilley et al., 2016), with the detection limit determined to be <1
pptV.
**2.3    Box Model**



We use the Dynamically Simple Model of Atmospheric Chemical Complexity (DSMACC) box
model (Emmerson and Evans, 2009) to interpret the observed $NO_x$ measurements. We focus on
the summer season (June, July, and August) as this has the largest data coverage. The model is
run for day 199 at 16.864°N, -024.868°W. We initialize the model with the mean observed $H_2O$,
CO, $O_3$, VOCs (Carpenter et al., 2010; Read et al., 2012), 100 $\mu m^2/cm^3$ aerosol surface area
(Carpenter et al., 2010). We also initialise the model with 1.5 ppt of $I_2$ and 2.5 ppt of $Br_2$ to
provide ~1.5 pptV of IO and ~2.5 pptV BrO during the day, consistent with the levels measured
over 9 months at the CVO during 2007 (Mahajan et al., 2010; Read et al., 2008). We use the
average diurnal cycle of the measured HONO concentrations, described above. Solar radiation at
this location in the tropics shows little seasonal variation, hence photolysis rates are similar in
summer and autumn. We assume clear sky conditions for photolysis. The unconstrained model is
run forwards in time until a stable diurnal cycle is attained; ~ 3 days. A full description of the
model chemistry is provided in the supplementary material. The base case chemistry has only
gas phase sources plus gas phase and deposition sinks for $NO_x$ as described in the supplementary
material.
**3       Results and discussion**
**3.1     Diurnal cycles in $NO_x$ and HONO**
Figure 1 shows the measured mean diurnal cycles of NO, $NO_2$, $NO_x$, and $O_3$ observed in each
season (Meteorological Spring – Mar, Apr, May; Summer – Jun, Jul, Aug; Autumn – Sep, Oct,
Nov; and Winter – Dec, Jan, Feb) during 2014 and 2015. Every season shows a strong diurnal
cycle in NO, peaking after solar noon at around ~13:00 to 14:00. The diurnal cycle of $NO_2$ is
much less pronounced but also exhibits weak maxima in the early afternoon. Overall this leads to
a maximum in $NO_x$ during the day. This behaviour is consistent throughout the year and air
mass, though not necessarily on a "day-to-day" basis.
The observed diurnal cycle in $NO_x$ is hard to explain with conventional chemistry. The increase
in night time $NO_x$ suggests a continuous source but the maximum around noon suggests a
photolytic source. Given the predominant $NO_x$ sink is reaction with OH to form $HNO_3$, it would
be expected that there would be a minimum in $NO_x$ during the day rather than a maximum.



Similar observations have been reported previously (Monks et al., 1998) at the Cape Grim Baseline Air Pollution station (-40.683, 144.670), a comparably remote site in the southern hemisphere, and during the Atlantic Stratocumulus Transition Experiment (ASTEX) cruise (~29°N, 24°W) which reported similar daytime $NO_x$ production (Carsey et al., 1997). The observed behaviour in the CVO $NO_x$ was historically attributed to thermal decomposition of $NO_y$ species (Lee et al., (2009).

Figure 2 shows the average diurnal cycle at CVO of measured HONO concentrations. The data exhibits a strong daytime maximum peaking at noon local time (Solar noon ~13:20) and reaching zero at night, indicating a photolytic source. HONO reaches zero at night through deposition, photolysis and reaction with OH suggesting no other surface source causing night time build-up as often is observed otherwise (Ren et al., 2010; VandenBoer et al., 2014; Zhou et al., 2002).

Daytime production of HONO is similarly hard to reconcile if its formation by the homogeneous OH + NO reaction (or other gas-phase $HO_x$-$NO_x$ chemistry, e.g. Li et al., (2014)). With NO mixing ratios below 5 pptV, OH measured peaking at ~ 0.25 pptV during the RHaMBLe campaign (Carpenter et al., 2010; Whalley et al., 2010) and a maximum noontime $j$HONO of 1.2 × $10^{-3}$ $s^{-1}$, a steady state HONO mixing ratio of ~ 0.04 pptV is found ($k_{(OH + NO)}$ = 7.4 ×$10^{-12}$ mol.cm$^{-3}$ $s^{-1}$). An additional source of HONO must be present to explain the observed concentrations.

Both the long-term $NO_x$ and the short-term HONO observations made at CVO are difficult to explain with purely gas phase chemistry. Both datasets show daytime maxima indicative of a photolytic source of either $NO_x$ or HONO, whereas gas phase chemistry would predict minima in $NO_x$ during daytime and two orders of magnitude less HONO.

### 3.2    Box modelling of $NO_x$ sources

Using the box model (section 2.3) we explore the observed diurnal variation in $NO_x$ and understand the role of different processes. Classically, the predominant source of $NO_x$ in remote regions is considered to be the thermal decomposition of compounds such as peroxyacetyl nitrate (PAN) which can be produced in regions of high $NO_x$ and transported long distances (Fischer et





al., 2014; Jacobi et al., 1999; Moxim et al., 1996). We consider a source of PAN which descends from the free troposphere and then thermally decomposes to $NO_2$ in the warm MBL. The main sink of $NO_x$ is conversion to $HNO_3$, which is slightly counterbalanced by a slow conversion of $HNO_3$ back into $NO_x$ through gas phase photolysis or reaction with OH. Figure 3 shows the model with a source of PAN which results in mixing ratios of 5 – 8 pptV, consistent with the few measurements made in the marine boundary layer, most notably by Jacobi et al., (1999) who measured levels from <5 to 22 pptV in the tropical Atlantic, and Lewis et al., (2007) who reported PAN mixing ratios of ~10 pptV in the remote mid-Atlantic MBL.

It is evident from the base case model results shown in Fig. 3 that the model fails to calculate the $NO_x$ diurnal cycle. Modelled $NO_x$ concentrations do increase during the night, consistent with the observations, but the model's minimum for $NO_x$ occurs during the day when the observations show a maximum. The modelled and measured HONO is also shown in Fig. 3, both peaking during midday with observations reaching 2.5 pptV whilst the model simulates only ~ 0.2 pptV. It is clear that long-range transport and thermal decomposition of $NO_y$ species such as PAN alone cannot explain the $NO_x$ diurnal at Cape Verde. A PAN-type continuous thermal decomposition forming $NO_x$ would be inconsistent with the diurnal maximum in $NO_x$ which is observed. The $NO_x$ source necessary to support a noon time maximum would have to show a strong day-time maximum to counter the strong diurnal in the sink.

This need for a diurnal cycle in the $NO_x$ source also suggests that the shipping source of $NO_x$ is unlikely to explain the diurnal cycle. The dominant source of ship $NO_x$ in the region occurs from the large container ships which pass the region on their way to South America or the Cape of Good Hope. It would appear unlikely that these emissions are systematically higher during the day than during the night and thus are unlikely to explain the observed diurnal signal.

There have been a number of papers which have identified much faster photolysis of nitrate within and on aerosol, than for gas phase nitric acid (Baergen and Donaldson, 2013; Cohan et al., 2008; Handley et al., 2007; Ndour et al., 2009; Scharko et al., 2014; Ye et al., 2016a, 2016b; Zhou et al., 2003). These studies have found that particulate nitrate photolysis rates can be up to ~3 orders of magnitude greater than gas phase $HNO_3$ photolysis in marine boundary layer





conditions (Ye et al., 2016b). There is also broad agreement between different studies on the
main photolysis product being nitrous acid (HONO) with $NO_2$ as a secondary species. The
product ratio appears dependent on aerosol pH (Scharko et al., 2014). This is shown in reaction
(R6) as particulate nitrate (p-$NO_3$) photolysing to HONO and $NO_2$ in a ratio $x$:$y$.
$p\text{-}NO_3 + h\nu \rightarrow x\text{HONO} + y\text{NO}_2$                                                                 (R6)
There is also evidence that the photolysis rate is positively correlated with relative humidity
(Baergen and Donaldson, 2013; Scharko et al., 2014). As such, particulate nitrate photolysis rates
appear to increase with increasing aerosol acidity and relative humidity. With the CVO site
experiencing relative humidity of 79 % on average (Carpenter et al., 2010) and aerosol
containing a significant acidic fraction (Fomba et al., 2014), particulate nitrate photolysis could
have a role to play in the $NO_x$ budget at Cape Verde.
In order to explore the implications for Cape Verde $NO_x$ chemistry, we re-ran the base model
removing the PAN source but including particulate nitrate (p-$NO_3$) photolysis (R7) leading to
HONO and $NO_2$ production, scaled to the gas phase photolysis of $HNO_3$. We use an aerosol
phase concentration of nitrate of 1.1 μg m$^{-3}$ (equivalent to 400 pptV), which is the mean
concentration found in PM10 aerosol at Cape Verde, with little apparent seasonal variability
(Fomba et al., 2014). The branching ratio of HONO to $NO_2$ production from reaction 6 ($x$ and $y$)
was set to 2:1 in line with the findings of Ye et al., (2016b). We scale the p-$NO_3$ photolysis rate
to gas phase $HNO_3$ photolysis by factors of 1, 10, 25, 50, 100, and 1000. The study of Ye et al.,
(2016b) describes enhancements of up to ~300 fold. The impact on the summer months is shown
in Fig. 4.
Including the photolysis of aerosol nitrate changes both the mean concentration and diurnal cycle
of $NO_x$ significantly. The diurnal $NO_x$ is now flat or peaks during the daytime, more consistent
with observations. We find the best approximation is achieved when the rate of particulate nitrate
photolysis is ~10 times that of $HNO_3$ which is broadly consistent with laboratory based
observations (Zhou et al., 2003). A wide variability of p-$NO_3$ photolysis rates on different
surfaces are reported (Laufs and Kleffmann, 2016; Ye et al., 2016a), thus the photolysis of
nitrate is uncertain and likely to be variable with aerosol composition. In all particulate nitrate





photolysis-only scenarios, depicted in Fig. 4 and Fig. 5, it is evident that p-NO$_3$ photolysis alone

is doesn't give the observed increase in night time NO$_x$ observations. Conversely the PAN only

scenario is insufficient to sustain daytime NO$_x$. It is therefore likely that the actual source of NO$_x$

is a combination of PAN entrainment from the free troposphere and particulate nitrate photolysis.

Combining the free-tropospheric source of PAN, and the photolysis of particulate nitrate at a rate

of 10 times the gas phase HNO$_3$ photolysis (Fig. 5) results in a model simulation with roughly

twice as much NO$_x$ both at night and during daylight but a roughly flat diurnal profile. Simulated

HONO peaks at local noon, similar to the observations.

### 3.3    NO$_x$ sinks

Figure 6 shows the rates of production and loss analysis for NO$_x$ in this simulation with both

PAN thermal decomposition and particulate nitrate photolysis. The largest net source of NO$_x$

after net sinks (such as halogen nitrate cycling) are removed is nitrate (NIT) photolysis to HONO

and NO$_2$. The major net sink is the formation of nitric acid by reaction of NO$_2$ with OH – though

the uptake of HNO$_3$ onto aerosol and subsequent rapid (compared to gas phase HNO$_3$) photolysis

acts to balance even this.

The pronounced drop in modelled NO$_2$ at sunrise is due to production of halogen nitrates

(XONO$_2$, X = I, Br) when HOX rapidly photolyses to produce XO which can then react with

NO$_2$ to produce XONO$_2$. XO is formed quickly and spikes in concentration leading to the rapid

loss of NO$_2$. This feature is not observed in the NO$_x$ observations during any season.

The diagnostics in Figure 6 show the role of the different sinks of NO$_x$. In that simulation these

are dominated by the gas phase reaction between NO$_2$ and OH but with the rapid formation and

subsequent hydrolysis of BrONO$_2$ and IONO$_2$  (R7) playing a major role (Sander et al., 1999).

The uptake coefficient (γ) of halogen nitrates onto aerosol therefore could have a strong

influence on the NO$_x$ diurnal.

XONO$_2$ + H$_2$O$_{(aer)}$ → HNO$_{3(aer)}$ + X$^+$ + OH$^-$                                                        (R7)





We perform a sensitivity analysis on the effect of the uptake coefficients on the $NO_x$ and XO
diurnals. We do this in a particulate nitrate photolysis only simulation, without PAN, to isolate
the effect of $XONO_2$ hydrolysis on nitrate-$NO_x$ cycling.  Figure 7 shows the effect of changing $\gamma$
of $XONO_2$ (X = Br, I) within recommended ranges (Burkholder et al., 2015; Saiz-Lopez et al.,
2008) on Saharan dust and sea salt – the predominant aerosol at Cape Verde, ranging from 0.02
to 0.8.
Increasing the $\gamma$ of $XONO_2$ from 0.02 (the low end of recommended values) to 0.1 results in
small changes to both the $NO_x$ and XO diurnals. The loss of $NO_x$ at sunrise becomes more
pronounced whereas the XO diurnals become slightly more 'square' or 'top-hat' as per the
observations of Read et al., (2008). Increasing the $\gamma$ to the upper extreme ($\gamma = 0.8$) results in a
spike in BrO at sunrise, which consumes the majority of $NO_2$ though formation of $BrONO_2$. No
combination of uptake coefficients can completely reproduce the characteristic XO diurnals.
The effect on the $NO_x$ diurnal of changing $\gamma$ is clear in that greater uptake coefficients
recommended by e.g. JPL (Burkholder et al., 2015) result in objectively worse simulation of both
the $NO_x$ and XO diurnals. It is therefore likely that information is lacking from the XO – $NO_x$
chemistry scheme as it is currently known.
### 3.4    HOI/HOBr - $NO_x$ chemistry
Recently, IO recycling by reaction with $NO_3$ has been proposed by Saiz-Lopez et al., (2016) who
calculated that the reaction (R8) of HOI + $NO_3$ producing IO and $NO_3$ has a low enough
activation energy and fast enough rate constant to be atmospherically relevant in the troposphere.
$HOI + NO_3 \rightarrow IO + HNO_3$    :        $k = 2.7 \times 10^{-12} (300/T)^{2.66}$                    (R8)
This mechanism provides a route to nitric acid, and thus particulate nitrate at night, whilst also
leading to nocturnal IO production leading to loss of $NO_2$ by $IONO_2$ formation.
Including this new reaction and re-running the model leads to a diurnal profile of IO much more
representative of the observations. This however introduces a more pronounced loss of $NO_x$ at



sunrise and sunset, and also results in $NO_x$ peaking during the day which fits better with the
observations as shown in Fig. 8.
The inclusion of this HOI + $NO_3$ reaction reproduces the general $NO_x$ and $O_3$ diurnals more
closely than without i.e. a daytime maximum in $NO_x$. There are also effects on the halogen oxide
behaviour. The simulated BrO has a flatter profile, which more closely matches the observations.
However, modelled IO is now non-zero at night and the sunrise build-up and sunset decay still
occurs more abruptly than the observations.
Although the $NO_x$ and $O_3$ diurnals are reproduced more closely with this new chemistry, there is
still disagreement with the observed $NO_x$ diurnal at sunrise and sunset especially indicating a
missing reaction or reactions. To best approximate the observed diurnal behaviour an analogous
HOBr + $NO_3$ night time reaction (R9) was introduced with a rate 10 times that of HOI + $NO_3$ as
calculated by Saiz-Lopez et al., (2016).
HOBr + $NO_3$ → BrO + $HNO_3$        :        $k = 2.7 \times 10^{-11} (300/T)^{2.66}$                    (R9)
This results in an improved reproduction of the observed $NO_x$ diurnal, Fig. 9. This is a purely
speculative representation in order to reproduce the observed $NO_x$ diurnal and highlights how
some mechanistic knowledge of $NO_x$-halogen-aerosol systems is still missing.
With HOX + $NO_3$ chemistry included in the model as in Fig. 9, significant loss of $NO_x$ at sunrise
and sunset is eliminated.  Greater HONO production is also simulated, with up to ~ 2.5 pptV
predicted – in line with the observations shown in Fig. 2. The improvement can be better
understood by diagnosing the modelled $NO_y$ distribution. In Fig. 10 the distribution of PAN,
$IONO_2$, $BrONO_2$, $N_2O_5$, and $NO_3$ is shown for the base case scenario (where entrained PAN is
the sole source of $NO_x$ in the MBL), for the particulate nitrate photolysis case including HOI +
$NO_3$ chemistry, and the same but also including HOBr + $NO_3$ chemistry. The major feature
changing through the different simulations is the magnitude and shape of the $BrONO_2$ diurnal.
From the base run (A) to the inclusion of HOI + $NO_3$ chemistry and particulate nitrate photolysis
(B) a major increase in $BrONO_2$ mixing ratio is expected at sun rise and sun set. It is this rapid
production of $BrONO_2$ which consumes $NO_x$ resulting in the sharp dips at these times not seen in





the observations. In the HOBr & HOI + $NO_3$ and particulate nitrate photolysis case (C) these features are eliminated and halogen nitrates do not spike at sunrise or sunset. This leads to a $NO_x$ diurnal which is more representative of the observations. Unsurprisingly, the inclusion of HOX + $NO_3$ chemistry results in lower mixing ratios of $NO_3$ at night. In all cases $N_2O_5$ (in black) is effectively zero at all times.

The agreement in modelled and observed $NO_x$ improves and the modelled values fall within the error of the observations. Additionally the approximate BrO diurnal is achieved – without the characteristic 'horns', however replicating IO observations is still problematic.

The effect of dramatically changing $NO_x$ diurnal could be expected to have an effect on OH and $HO_2$ mixing ratios. The difference between the base model case, where PAN decomposition is the dominant daytime source, and the final model where the $NO_x$ is more accurately described by particulate nitrate photolysis and HOX + $NO_3$ chemistry is shown in Fig. 11.

In the case of OH the change from the base model to the final model is an increase of 3.3% at the maximum, for $HO_2$ the increase is a more significant 8.6% (or 1.7 pptV), however this is well within the uncertainty of measured values (Whalley et al., 2010). Figure 11 shows that even with dramatic changes in the $NO_x$ simulation, the OH and $HO_2$ changes very little comparatively despite increased daytime HONO production.

From these simulations it would appear that the photolysis of aerosol phase nitrate may be the dominant source of $NO_x$ into the marine boundary layer around Cape Verde. Particulate nitrate photolysis would be capable of producing a diurnal cycle in $NO_x$ which was consistent with the observations when HOX + $NO_3$ chemistry is considered also. Whilst agreement between model and observation is improved there is a clear gap in understanding the halogen-$NO_x$-aerosol system in the remote marine boundary layer.

## 4    Conclusions

Fast aerosol nitrate photolysis is shown to be likely the primary source of $NO_x$ in the remote tropical Atlantic boundary layer. A 0-D model replicated the observed halogen, $O_3$, OH, $NO_x$ and HONO levels when including particulate nitrate photolysis at a rate of ~10 times that of gas





phase nitric acid, consistent with previous laboratory measurements. Model optimisation shows that this new source of daytime $NO_2$ is compatible with observations and currently known chemistry at night and at mid-day, but that at sunrise and sunset there is disagreement due to the treatment of halogen oxides at these times. Recently suggested halogen hydroxide + nitrate radical chemistry may provide better agreement between model and observation when theoretical reactions can be substantiated.

**Data Availability**

We thank the NASA Jet Propulsion Laboratory (Burkholder et al., 2015) for providing comprehensive rate and uptake coefficient data for atmospheric compounds, which can be found at http://jpldataeval.jpl.nasa.gov.

All data used in this work is available from the British Atmospheric Data Centre (BADC) http://badc.nerc.ac.uk and is included as a .csv file in the supplementary information. The DSMACC model is available from https://github.com/barronh/DSMACC and a full description of the model can be found in the supplementary information.

*Acknowledgements.* The authors would like to thank Luis Neves Mendes of the Instituto Nacional de Meteorologia e Geofísica (INMG) for their operational support at the CVO site. The financial support of NCAS, the National Centre for Atmospheric Science, and of NERC, the Natural Environmental Research Council for supporting the studentship of Chris Reed is gratefully acknowledged. HONO measurements were support by NERC grant NE/M013545/1 (Sources of Nitrous Acid in the Atmospheric Boundary Layer).

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



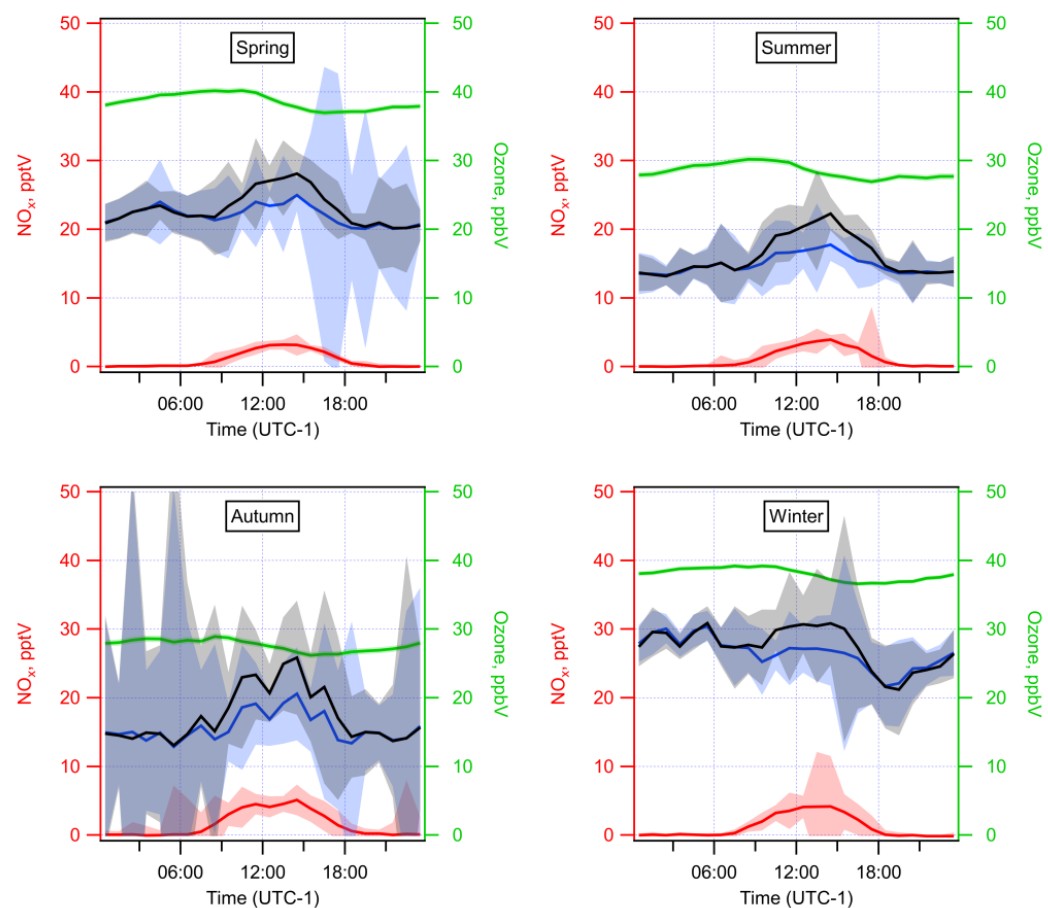

3    Figure 1. The observed seasonal diurnal cycles in NO, $NO_2$, $NO_x$, and $O_3$ at the CVO GAW

4    station during 2014 and 2015. NO is shown in red, $NO_2$ in blue, $NO_x$ in black, and $O_3$ in green.

5    Shaded areas indicate the standard error of data.





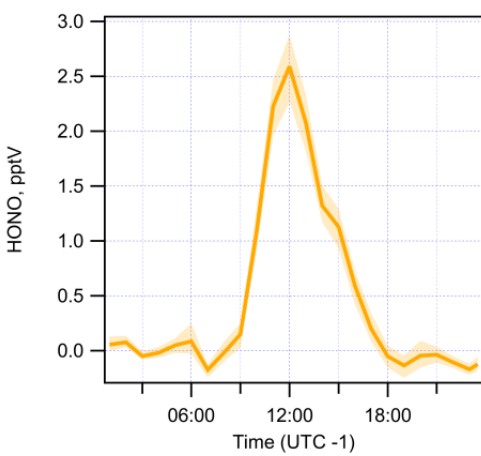

2   Figure 2. The observed average HONO diurnal measured at CVO during 24[th] November – 3[rd]

3   December 2015. Shaded area indicates standard deviation of data.



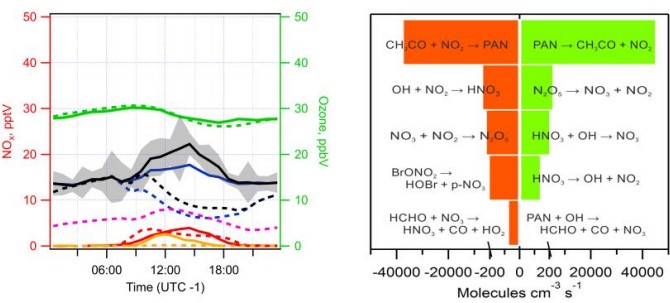

Figure 3. Left shows the measured (solid lines) and modelled (dashed) $NO_x$ and HONO diurnal
behaviour at the CVO GAW station where the dominant source of $NO_x$ is a source of PAN
descending from the upper troposphere having been transported from polluted regions. $O_3$ –
green; $NO_x$ – black; $NO_2$ – blue; NO – red; HONO – yellow; PAN – pink.  Right shows the rates
of production and loss of NO and $NO_2$ from sources listed in descending order of contribution
over a 24 hour period accounting for >95% of the total. Shaded areas are standard error of the
observation.





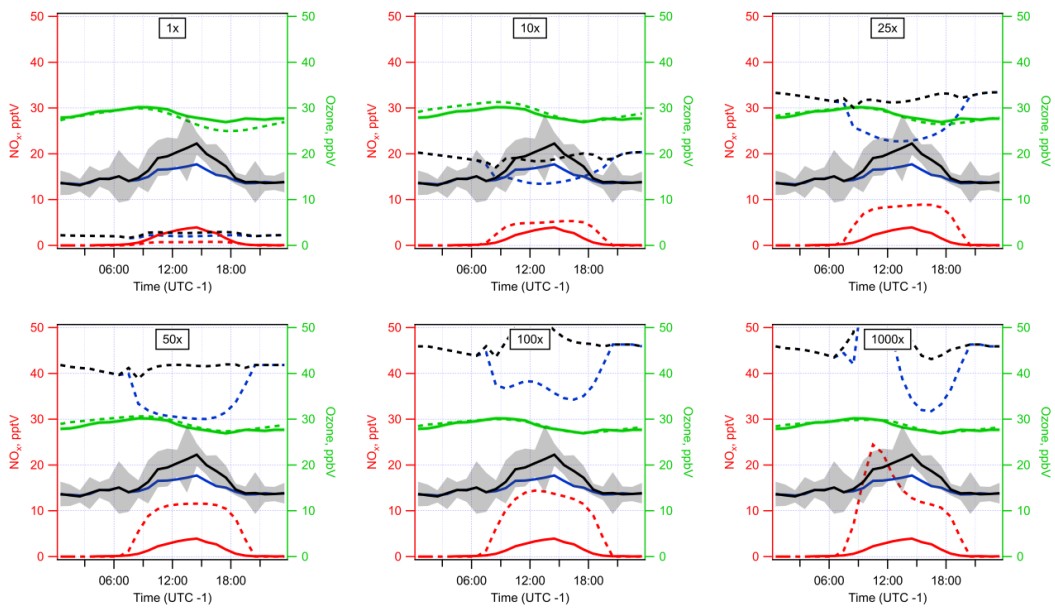

Figure 4. The modelled diurnal profile of $NO_x$ at CVO during summer months when photolysis
of nitrate is considered. The rate of particulate nitrate photolysis has been scaled to the rate of
$HNO_3$ photolysis by factors of 1, 10, 25, 50, 100, and 1000. Observations are solid lines whilst
modelled values are shown dashed. Shaded areas are standard error of the observation. $O_3$ –
green; $NO_x$ – black; $NO_2$ – blue; NO – red.



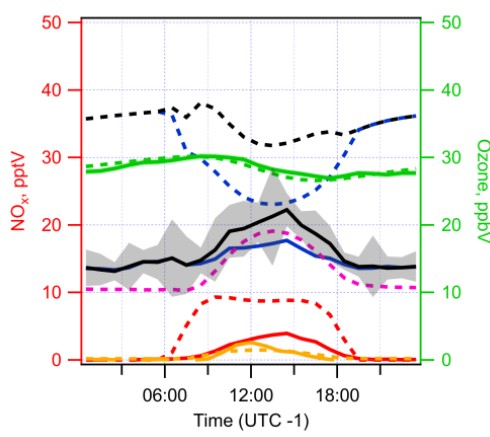

Figure 5. The modelled diurnal profile of $NO_x$ at CVO during summer months when photolysis
of nitrate (set at 10× the gas phase $HNO_3$ photolysis) and a tropospheric PAN source are
considered. Shaded areas for $NO_x$ are the standard error of the observation. $O_3$ – green; $NO_x$ –
black; $NO_2$ – blue; NO – red; HONO – yellow; PAN – pink.





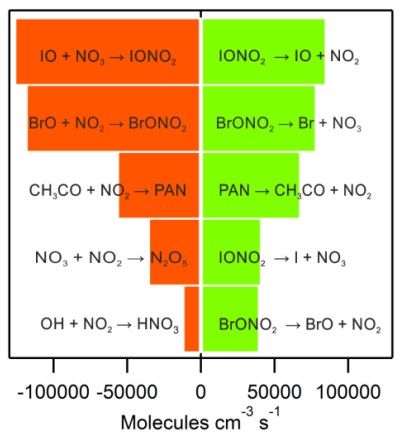 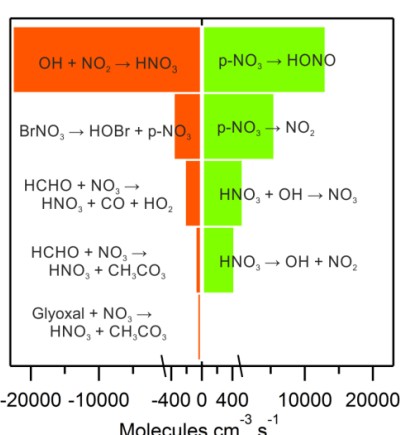

2  Figure 6. Left is the production and loss analysis of the combined model of particulate nitrate

3  photolysis and PAN decomposition over 24 hours. Right is the same analysis discarding the

4  major balanced sinks of fast cycling short lived species.





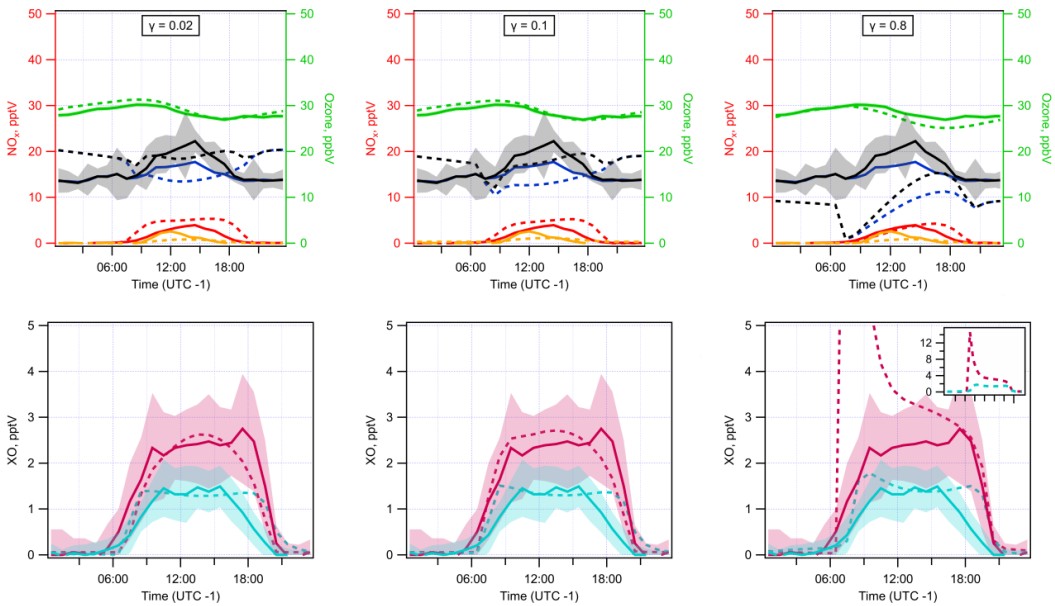

Figure 7. Sensitivity analysis of the effect of changing reactive uptake co-efficients (γ) of
reactive halogens (XO, XHO, $XONO_2$, X = Br, I) on $NO_x$ (top) and XO (bottom) diurnal
behaviour during summer months at CVO. Particulate nitrate photolysis is set at 10 times the rate
of gaseous $HNO_3$. Observations are solid lines whilst modelled values are shown as dashed. IO
and BrO observations are adapted from Read et al., (2008). Shaded areas are standard error of
the observation. $O_3$ – green; $NO_x$ – black; $NO_2$ – blue; NO – red; HONO – yellow; PAN – pink;
IO – turquoise; BrO – purple.





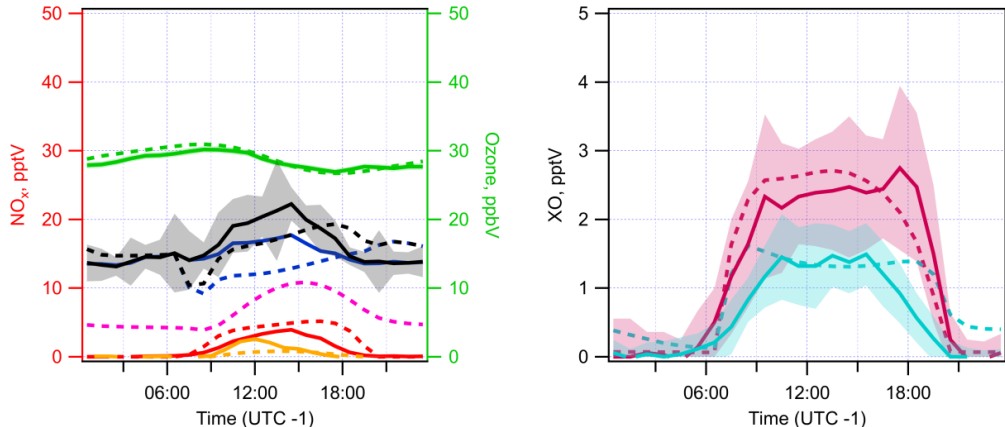

Figure 8. Left is the modelled $NO_x$ and HONO diurnal cycle for the CVO site during summer
months with the inclusion of night time HOI chemistry. Centre is the observed (adapted from
Read et al., (2008)) and modelled IO and BrO. Observations are solid lines whilst modelled
values are shown dashed. Shaded areas are standard error of the observation. $O_3$ – green; $NO_x$ –
black; $NO_2$ – blue; NO – red; HONO – yellow; PAN – pink; IO – turquoise; BrO – purple.





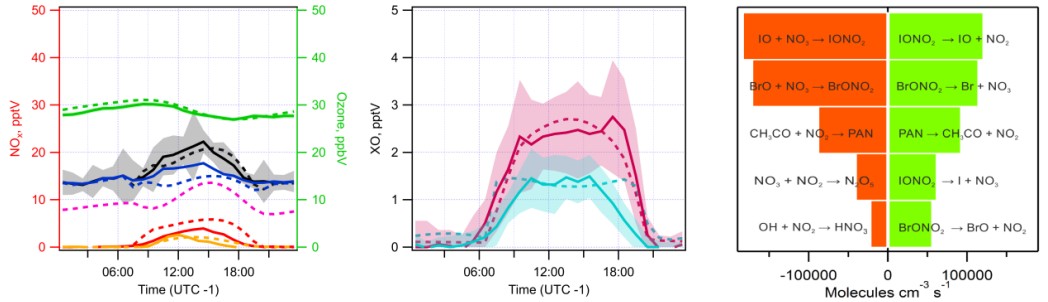

Figure 9. $NO_x$ and halogen oxide diurnals for the CVO site during summer months. Observations
are solid lines (BrO and IO adapted from Read et al., (2008)) whilst modelled values are shown
dashed. Shaded areas are standard error of the observation. $O_3$ – green; $NO_x$ – black; $NO_2$ – blue;
NO – red; HONO – yellow; PAN – pink; IO – turquoise; BrO – purple. Night-time $HOI + NO_3$
chemistry is included as is speculative HOBr chemistry analogous to that of HOI.





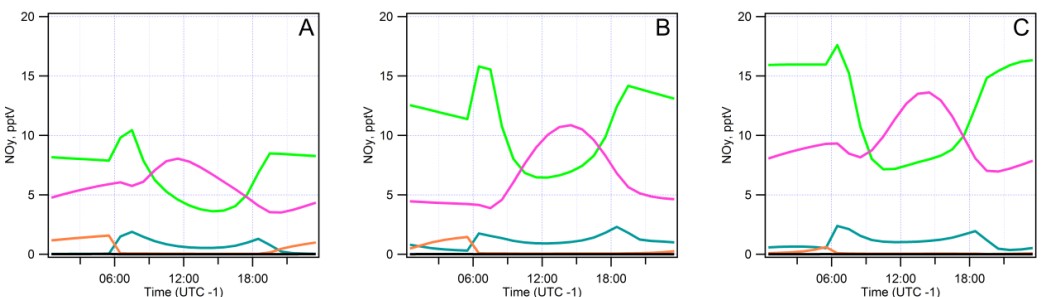

2    Figure 10. Shown are $NO_y$ diurnals for the CVO site during summer months in the base scenario

3    (A), with $HOI + NO_3$ chemistry included (B), and with $HOI$ & $HOBr + NO_3$ chemistry included

4    (C). $BrONO_2$ = green, $IONO_2$ = teal, PAN = pink, $NO_3$ = orange, $N_2O_5$ = black.





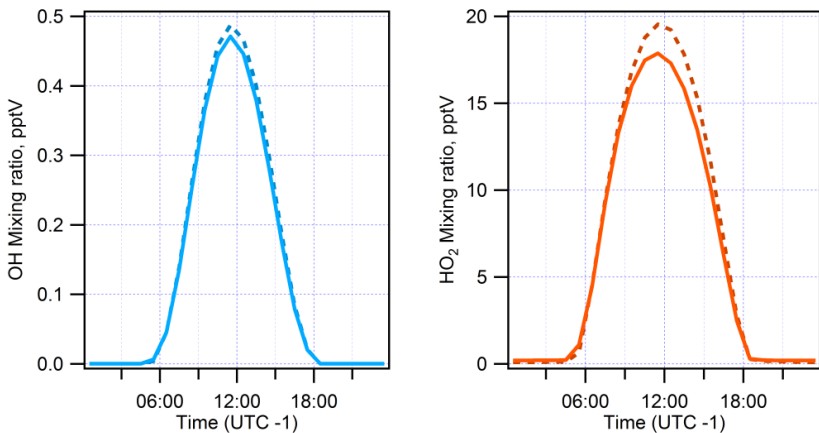

Figure 11. Modelled OH (left) and $HO_2$ (right) mixing ratios comparing the base case model
where PAN decomposition is the dominant source of $NO_x$ in the remote MBL (solid lines), with
the final model where the dominant source of $NO_x$ is particulate nitrate photolysis and HOX +
$NO_3$ chemsitry is included.

