# Peer review of "Evidence for renoxification in the tropical marine boundary layer"

_Atmospheric Chemistry and Physics, 2016_

## Referee Comment (RC1) · Anonymous Referee #1 · 31 Jan 2017

General comments:

The Authors present a compelling set of model results to explain the chemistry underpinning commonly observed daytime maxima in NOx at the Cape Verde Atmospheric Observatory. The impact of condensed phase nitrate photolysis in improving understanding of the NOx cycle in the marine boundary layer from long-term datasets, which capture diurnal features, has not been presented to date. This manuscript provides a robust method with which to test the findings of intensive field and lab observations of this phenomenon. The authors also find that the role of halogens is important in describing several features in the temporal nature of the NOx diurnal patterns, building on recent findings that such chemistry may be important in controlling the cycling of reactive nitrogen in remote marine regions. The presented manuscript is well-written and most of the data in the figures is presented clearly. Overall, this work is acceptable

for publication in Atmospheric Chemistry and Physics after a number of minor revisions and technical corrections have been made, which are presented in detail below.

Minor comments:

1) The title of the manuscript does not convey two of the major topics of this paper, nitrous acid and halogen hydroxides. The authors should consider modifying their title to reflect the important roles of HONO and halogens on renoxification processes in this work.

2) The authors cite a number of real-world surface (Baergen and Donaldson, 2013; Ye et al., 2016a), laboratory substrate studies (Handley et al., 2007; Scharko et al., 2014; Zhou et al., 2003), a model estimate (Cohan et al., 2008) and two works on aerosol nitrate catalytic degradation (Ndour et al., 2009) and photolysis (Ye et al., 2016b) as the basis for parameterizing the particulate nitrate conversion rates in their model (e.g Pages 7-8). The majority of the cited work is for nitrate photolysis on proxy surfaces at the atmospheric interface and this is not clearly stated throughout the manuscript, which makes the focus on aerosol nitrate photolysis throughout the manuscript some-what confusing. The connection and implications of linking nitrate photolysis on/in these other condensed phases is not clear in its applicability or in its limitations and this would be worth expanding on in the manuscript. This photochemistry is obviously important in this environment, but if studies of surface media and bulk aqueous solution (Scharko et al., 2014) are used to constrain the rates in the model and contribute to the discussion (e.g. effects of pH and RH), then the discussion should be expanded to include the expected role of surfaces versus aerosols in the MBL or how the parame-terization of aerosol photolysis encompasses all of these sources.

3) The rapid photolysis of aerosol nitrate suggests that the lifetime of the reservoir may be quite short during the day (∼hours, (Ye et al., 2016b)), but this may be dependent on the chosen photolysis rate. It would be worthwhile to discuss this and present the diurnal trend from the model (or indicate that this term is held constant) to compare to

these previous findings. Clarifying whether the aerosol nitrate photolysis mechanism can operate on the ambient aerosol observed without depleting it and discussing how the reservoir is maintained would improve the argument that this is a reasonable HONO and NOx source.

4) The (Crilley et al., 2016) manuscript only cites a (Heland et al., 2001) paper on the LOPAP, without any operational details on how such low detection limits were achieved for the instrument used in this work. The majority of the data in Figure 2 are below the stated LOD of the LOPAP (< 1pptv), suggesting that this data has an associated high uncertainty, which is not depicted. What is the exact LOD of the instrument and what data can be reliably reported in this figure? It would also be helpful to present the methods for calibration, background correction, and determining the precision and accuracy of the measurements, as those achieved here are non-trivial.

5) The Authors focus their model on the 'summer season' (Page 5, Line 3) as this is the period of greatest data density from CVO. Is this dataset also filtered for clear-sky days to reduce comparison bias between the model and measurements? This season is most likely to be affected by cloudy days according to CVO observations reported in (Carpenter et al., 2010). Also, the model compares to HONO measurements from the winter period, when ambient NOx measurements do not exhibit the same diurnal pattern (i.e. the mid-day maximum) that the majority of this manuscript seeks to explain. It would be useful to present why the authors expect that winter HONO mixing ratios and diurnal structure are representative of summer HONO. Finally, how have the authors included or reasonably excluded boundary layer and transport dynamics in their 0D model? The report from (Carpenter et al., 2010) states that the limited information available in this regard indicates no diel modulation of the boundary layer height, but that it can change substantially from day to day at a site 200 km away. The work cited for DSMACC (Emmerson and Evans, 2009) does not suggest how the boundary layer is represented in the model and it could be that some of the mismatch in early morning and evening NOx levels is due to mixing and entrainment or local transport phenomena

instead of the model chemistry. It would be useful for the authors to discuss such processes as being accounted for, or as limitations, in the methods and at the appropriate points in the manuscript discussion (e.g. (Wolfe et al., 2016)).

Technical corrections:

Page 1, Line 26: 'the box model simulation' of what? Everything?

Page 2, Line 3: provide a range of typical values for NOx observations in the remote MBL

Page 2, Lines 13-16: This is an example where the authors specify only particulate nitrate, yet cite work probing a variety of condensed phase nitrate proxies, ranging from surface-adsorbed nitrate to bulk aqueous solutions. The authors should be more specific here regarding the media and interfaces (e.g. particle-gas, surface-gas, aqueous-gas) these works have described and that they have all found an enhancement in nitrate photolysis in the condensed phase although the mechanisms are not well understood.

Page 2, Line 17: Delete 'through'

Page 2, Lines 17-21: Specify that these reactions are all taking place in the gas phase.

Page 3, Line 17: 'for a short period in Nov/Dec 2015' should be restated to the number of days in the winter of 2015, with the observational period explicitly given in the HONO measurement section.

Page 3, Line 23 - Page 4, Line 4: Is assessment of RH and O3 effects on NO sensitivity and NO2 converter efficiency at a period of 71 hours assuming that there is little change in sample RH and O3 over time or that the 1 hour offset in performing this calibration, spread over 2 years, corrects for these diurnal variations over the long term? Also, the measurements reported by (Lee et al., 2009) indicate that this period was 37 hours long. The authors describe in detail how RH of the sample flow is minimized, but do not present information as to the range or relative proportion of RH that sample flows

are reduced to/by. O3 has a clear diurnal cycle presented throughout the manuscript, so it would be expected that corrections are necessary on an hourly timescale, not once every three days. While many other interferences are clearly detailed for the approach to correction in (Lee et al., 2009; Reed et al., 2016a, 2016b) this particular modification and approach could be more clearly demonstrated to have little variation, with an example of the variability that would be required to have significant systematic bias in the measurement due to RH and O3 changes within a 71 hour period.

Also, the authors do not present any information about whether aerosols are removed from the sample flow, which could lead to artifact NOx signals in the system, similarly to other adsorbed species in the photolysis cell. A greater description of the main lab manifold at the beginning of this section would be sufficient to clarify.

Page 4, Line 15: Should 'period' be 'range'? 'being' should be 'are' and since much of this section is reporting data to two significant digits, shouldn't the detection limit for NO of 0.3 be 0.30?

Page 4, Line 21: The 'main lab manifold' is not described above. It would be very useful to have this presented above to know how external air is being delivered to the NOx instrumentation.

Page 4, Lines 25-26: Do the PM measurements at the site ever indicate the presence of nitrite? Given the prevalence of dust impacting the site, nitrite could be formed on these surfaces. The LOPAP has been shown to effectively sample large aerosols, such as fog droplets, and the authors state dust and sea salt as dominating the mass transport of condensed nitrate to CVO. This could bias the HONO measurement as LOPAP instrumentation does not typically exclude such coarse particles (e.g. (Sörgel et al., 2011)) and the dual-channel scrubbing coils used to quantify background and interference signals only effectively transmit particles less than 1 micrometer in diameter.

Page 6, Line 6: should this sentence finish 'in the instrument inlet'? From (Reed et al., 2016a, 2016b) the thermal decomposition of PAN seems to occur in the photolysis

cell?

Page 6, Lines 10-11: This sentence is describing nocturnal processes, yet photolysis and OH losses are listed. Please correct this error. Also, there is evidence in the presented data that the HONO buildup at night is still occurring (data below 0 pptV at 18:00, and above at 06:00) as would be expected, from the measured precursor NO2 being present at night and able to undergo heterogeneous hydrolysis. This may not be statistically significant, depending on the uncertainty in the HONO measurement, or the data may only be an estimate based on the exact instrument detection limits, so some clarification here should be given by considering those two limits. Previous work has also shown a rapid approach to steady state in nocturnal HONO in marine environments due to reversible thermodynamic partitioning in marine boundary layer surface waters, which is not mentioned here (Wojtal et al., 2011).

Page 6, Line 17: 'daytime' should be placed between 'additional' and 'source'

Page 6, Lines 19-20: 'are difficult to explain' should be 'cannot be explained'

Page 6, Line 21: 'either of NOx or HONO'. Shouldn't this be 'NOx and HONO'?

Page 7, Lines 22-23: The authors should replace 'would appear' with 'is'. Also, it would seem that the intrusion of ship emissions, if stochastic, would be normalized from the mean through the consideration of 2 years of summer data. This is supported by the range versus the mean of the NOx data presented in many figures.

Page 8, Lines 14-17: It would be expected that the aerosol nitrate would be distributed across both fine and coarse mode aerosol and photolyze differently based on their optical and chemical properties. The authors state in lines 26-28 that this is the case. It would be useful to clarify that the best match of nitrate photolysis enhancement that reproduces observed HONO is a rate integrated across all surface nitrate photolysis sources at CVO since only bulk aerosol composition has been measured in (Fomba et al., 2014).

Page 8, Lines 26-28: It is confusing why the authors cite the (Laufs and Kleffmann, 2016) work here as they state in the abstract of their work, a conclusion counter to the thesis of this work:

'If these results can be translated to atmospheric surfaces, HNO3 photolysis cannot explain the significant HONO levels in the daytime atmosphere. In addition, it is demonstrated that even the small measured yields of HONO did not result from the direct photolysis of HNO3 but rather from the consecutive heterogeneous conversion of the primary photolysis product NO2 on the humid surfaces. The secondary NO2 conversion was not photoenhanced on pure quartz glass surfaces in good agreement with former studies. A photolysis frequency for the primary reaction product NO2 of J(HNO3 - NO2) = 1.1x10ˆ-6 sˆ-1 has been calculated (0 SZA, 50% r.h.), which indicates that renoxification by photolysis of adsorbed HNO3 on non-reactive surfaces is also a minor process in the atmosphere.'

The work described by the cited works of (Baergen and Donaldson, 2013, 2016; Scharko et al., 2014; Ye et al., 2016a, 2016b; Zhou et al., 2003) are all in disagreement with (Laufs and Kleffmann, 2016) and the photolysis rates from these measurements are used to constrain this model. They also clearly discuss the wide range of photolysis values without such contradictory statements. The authors should consider revising the works cited in this location.

Page 9, Line 1: Figure 5 includes PAN transport. Remove the reference to it here.

Page 9, Lines 5-8: The ability to reproduce the NOx profile is based on a large loss of NO2 and production of NO, the former of which is not observationally consistent. Stating this and the need to explore further chemical mechanistic constraints would improve the transition to the next section of the manuscript.

Page 9, Line 9: It would be useful to include some reference to halogen chemistry in this section header

[Figure]

Page 9, Line 12: '(NIT)' this is the only instance of this shorthand in the manuscript. Delete.

Page 9, Lines 13-15: This would be much easier to follow if broken into 2-3 sentences.

Page 10, Lines 5-6: Dust and sea salt are stated to be the 'predominant aerosol' at CVO. Is this by number, mass, or surface area? Please specify, with reference to (Carpenter et al., 2010; Fomba et al., 2014), so there is greater clarity in understanding if the majority of the nitrate is expected to be in the coarse mode.

Page 10, Line 14: Delete 'e.g. JPL' and change the citation format to 'Burkholder et al., (2015)'

Page 10, Lines 15-16: It would seem that the heterogeneous chemistry on fine mode aerosol may be what is poorly constrained. Would it be possible to speculate on this?

Page 10, Line 19: 'NO3' should be 'HNO3'. Also, is the static reactive uptake coefficient of 0.15 used in the model for HNO3 partitioning reasonable given the likely need for this value to increase mid-day to sustain the reservoir of particulate nitrate?

Page 12, Lines 8-9: This seems like a transition to an 'Atmospheric Implications' section

Page 12, Line 18: Update this to include the role of other surfaces.

References: Chemical subscripts and capitalization issues need to be corrected in: Burkholder et al (2015), Evans and Jacob (2005), Handley et al (2007), Laufs and Kleffmann (2016), Li et al (2014), Moxim et al (1996), Nakamura et al (2003), Pollack et al (2011), Ryerson et al (2000), Saiz-Lopez et al (2008), Sander et al (1999), Scharko et al (2014), Ye et al (2016a), and Zhou et al (2003)

Figure 1: Why is the NOx axis red, when the NOx trace is black? The color scheme here is generally not suitable for red-green color blind individuals and also does not print well in grayscale. Consider a scheme for figures, to use throughout, that is more

[Figure]

easily discerned.

Standard error is weighted by the number of samples considered, but those values are not presented anywhere. It would be worthwhile to do so, especially for the summer period.

The rest of the manuscript only considers the summer observations. Thus, only 'summer' requires a definition of the months considered. Labels in the figure could just be the months considered and would remove the need to cross-reference.

Figure 2: Add the cumulative accuracy and precision error and depict the instrument detection limit.

Figure 3: (left) For all plots like this, would it be more informative to present the values of the difference between the model and the measurement? The color and formatting challenges noted in Figure 1 apply here too. (right) The reaction text is difficult to read and the scale breaks are confusing. Would a log scale work and still emphasize the necessary rates?

Figure 4: This figure could be simplified if the difference between NOx, NO2, and NO relative to the observations were depicted in three separate panels for the photolysis factors considered. It would also be a more quantitative representation of which factor is most suitable.

Figure 6: Can the magnitude of the particulate nitrate photolysis be presented here? It would be nice to compare it to the other NOx source mechanisms. Also, it is surprising that HONO photolysis isn't presented as the manuscript suggests that its intermediate nature is key in reNOxification at CVO. (right) Same comments as Fig 3. (caption) Insert 'for NOx' after 'loss analysis'

Figure 7: There is no PAN on this figure, but it is listed in the caption. The difference notation, again, may be more informative for presenting the comparisons.

Figure 8: Could the dips in the early morning NOx in the model be mismatching the

observations because of NOx transport or dilution that isn't accounted for in the 0D model?

Figure 11: What do the dashed lines represent?

References

Baergen, A. M. and Donaldson, D. J.: Photochemical renoxification of nitric acid on real urban grime, Environ. Sci. Technol., 47(2), 815–820, doi:10.1021/es3037862, 2013.

Baergen, A. M. and Donaldson, D. J.: Formation of reactive nitrogen oxides from urban grime photochemistry, Atmos. Chem. Phys., 16(10), 6355–6363, doi:10.5194/acp-16-6355-2016, 2016.

Carpenter, L. J., Fleming, Z. L., Read, K. A., Lee, J. D., Moller, S. J., Hopkins, J. R., Purvis, R. M., Lewis, A. C., Müller, K., Heinold, B., Herrmann, H., Fomba, K. W., Van Pinxteren, D., Müller, C., Tegen, I., Wiedensohler, A., Müller, T., Niedermeier, N., Achterberg, E. P., Patey, M. D., Kozlova, E. A., Heimann, M., Heard, D. E., Plane, J. M. C., Mahajan, A., Oetjen, H., Ingham, T., Stone, D., Whalley, L. K., Evans, M. J., Pilling, M. J., Leigh, R. J., Monks, P. S., Karunaharan, A., Vaughan, S., Arnold, S. R., Tschritter, J., Pöhler, D., FrieÃ, U., Holla, R., Mendes, L. M., Lopez, H., Faria, B., Manning, A. J. and Wallace, D. W. R.: Seasonal characteristics of tropical marine boundary layer air measured at the cape verde atmospheric observatory, J. Atmos. Chem., 67(2–3), 87–140, doi:10.1007/s10874-011-9206-1, 2010.

Cohan, A., Chang, W., Carreras-Sospedra, M. and Dabdub, D.: Influence of sea-salt activated chlorine and surface-mediated renoxification on the weekend effect in the South Coast Air Basin of California, Atmos. Environ., 42(13), 3115–3129, doi:10.1016/j.atmosenv.2007.11.046, 2008.

Crilley, L. R., Kramer, L., Pope, F. D., Whalley, L. K., Cryer, D. R., Heard, D. E., Lee, J. D., Reed, C. and Bloss, W. J.: On the interpretation of in situ HONO observations via photochemical steady state, Faraday Discuss., 189(0), 191–212,

doi:10.1039/C5FD00224A, 2016.

Emmerson, K. M. and Evans, M. J.: Comparison of tropospheric gas-phase chemistry schemes for use within global models, Atmos. Chem. Phys., (1990), 1831–1845, doi:10.5194/acpd-8-19957-2008, 2009.

Fomba, K. W., Müller, K., Van Pinxteren, D., Poulain, L., Van Pinxteren, M. and Herrmann, H.: Long-term chemical characterization of tropical and marine aerosols at the Cape Verde Atmospheric Observatory (CVAO) from 2007 to 2011, Atmos. Chem. Phys., 14(17), 8883–8904, doi:10.5194/acp-14-8883-2014, 2014.

Handley, S. R., Clifford, D. and Donaldson, D. J.: Photochemical loss of nitric acid on organic films: A possible recycling mechanism for NOX, Environ. Sci. Technol., 41(11), 3898–3903, doi:10.1021/es062044z, 2007.

Heland, J., Kleffmann, J., Kurtenbach, R. and Wiesen, P.: A new instrument to measure gaseous nitrous acid (HONO) in the atmosphere, Environ. Sci. Technol., 35(15), 3207–3212, doi:10.1021/es000303t, 2001. Laufs, S. and Kleffmann, J.: Investigations on HONO formation from photolysis of adsorbed HNO3 on quartz glass surfaces, Phys. Chem. Chem. Phys., 18(14), 9616–9625, doi:10.1039/c6cp00436a, 2016.

Lee, J. D., Moller, S. J., Read, K. A., Lewis, A. C., Mendes, L. and Carpenter, L. J.: Year-round measurements of nitrogen oxides and ozone in the tropical North Atlantic marine boundary layer, J. Geophys. Res. Atmos., 114(21), 1–14, doi:10.1029/2009JD011878, 2009.

Ndour, M., Conchon, P., D'Anna, B., Ka, O. and George, C.: Photochemistry of mineral dust surface as a potential atmospheric renoxification process, Geophys. Res. Lett., 36(5), 2–5, doi:10.1029/2008GL036662, 2009.

Reed, C., Brumby, C. A., Crilley, L. R., Kramer, L. J., Bloss, W. J., Seakins, P. W., Lee, J. D. and Carpenter, L. J.: HONO Measurement by Differential Photolysis, Atmos. Meas. Tech. Discuss., (2), 1–28, doi:10.5194/amt-2016-17, 2016a.

Reed, C., Evans, M. J., Di Carlo, P., Lee, J. D. and Carpenter, L. J.: Interferences in photolytic NO2 measurements: Explanation for an apparent missing oxidant?, Atmos. Chem. Phys., 16(7), 4707–4724, doi:10.5194/acp-16-4707-2016, 2016b.

Scharko, N. K., Berke, A. E. and Raff, J. D.: Release of Nitrous Acid and Nitrogen Dioxide from Nitrate Photolysis in Acidic Aqueous Solutions, Environ. Sci. Technol., doi:10.1021/es503088x, 2014.

Sörgel, M., Trebs, I., Serafimovich, A., Moravek, A., Held, A. and Zetzsch, C.: Simultaneous HONO measurements in and above a forest canopy: Influence of turbulent exchange on mixing ratio differences, Atmos. Chem. Phys., 11(2), 841–855, doi:10.5194/acp-11-841-2011, 2011.

Wojtal, P., Halla, J. D. and McLaren, R.: Pseudo steady states of HONO measured in the nocturnal marine boundary layer: A conceptual model for HONO formation on aqueous surfaces, Atmos. Chem. Phys., 11(7), 3243–3261, doi:10.5194/acp-11-3243-2011, 2011. Wolfe, G. M., Marvin, M. R., Roberts, S. J., Travis, K. R. and Liao, J.: The framework for 0-D atmospheric modeling (F0AM) v3.1, Geosci. Model Dev., 9(9), 3309–3319, doi:10.5194/gmd-9-3309-2016, 2016.

Ye, C., Gao, H., Zhang, N. and Zhou, X.: Photolysis of Nitric Acid and Nitrate on Natural and Artificial Surfaces, Environ. Sci. Technol., 50(7), 3530–3536, doi:10.1021/acs.est.5b05032, 2016a.

Ye, C., Zhou, X., Pu, D., Stutz, J., Festa, J., Spolaor, M., Tsai, C., Cantrell, C., Mauldin, R. L., Campos, T., Weinheimer, A., Hornbrook, R. S., Apel, E. C., Guenther, A., Kaser, L., Yuan, B., Karl, T., Haggerty, J., Hall, S., Ullmann, K., Smith, J. N., Ortega, J. and Knote, C.: Rapid cycling of reactive nitrogen in the marine boundary layer., Nature, 532(7600), doi:10.1038/nature17195, 2016b.

Zhou, X., Gao, H., He, Y., Huang, G., Bertman, S. B., Civerolo, K. and Schwab, J.: Nitric acid photolysis on surfaces in low-NO x environments: Significant atmospheric

implications, Geophys. Res. Lett., 30(23), 2217, doi:10.1029/2003GL018620, 2003.

---

## Referee Comment (RC2) · Anonymous Referee #2 · 1 Mar 2017

Overall Assessment

The manuscript by Reed et al. presents 2-years of results on a unique diurnal cycle of NO, NO2 and O3 concentrations in the marine boundary layer from measurements at a coastal site in Cape Verde Atmospheric Observatory (CVO). Of particular interest is a noon-time high in NOx concentrations. A box model approach was used to model this diurnal profile, which is explained as arising from particulate nitrate photolysis and reactions of reactions of halogen nitrates (products of nitrate radical and halogen hydroxide chemistry during the nighttime). The authors argue that field observations could not be explained by dissociation of PAN that is transported to the site from anthropogenic sources over long distances. The methodology associated with measurements of NOx, HONO, O3 etc. are appropriately chosen and carefully executed. I think there is a potentially interesting data set here and a nice opportunity to explore the role of nitrate

aerosol photochemistry as a daytime NOx source. However, for completeness I would like to see an analysis of the relative importance of ClNO2 as a daytime NOx source vs. the other potential daytime NOx sources that were postulated.

The authors mention that the diurnal pattern in the CVO NOx concentrations was historically attributed to thermal decomposition of NOy species (see p. 6, L6). By NOy, the authors refer mostly to PAN, but what about N2O5 heterogeneous chemistry? Consideration of N2O5 heterogeneous chemistry appears to be limited to hydrolysis (modeled using N2O5==>2 NO3-, with an uptake coefficient of 0.02). This likely explains why in Fig. 10 the model shows non-existent N2O5 concentrations at this site over a 24 hour period. However, previous studies of N2O5 in coastal regions show that steady-state concentrations of 20-100 ppt can exist, with peaks during the nighttime. Those studies also demonstrate that aside from hydrolysis to form particulate nitrate, a major fate for N2O5 is conversion to ClNO2 on sea salt aerosol and the ocean surface. (e.g. PNAS, 2014, 111, 3943). Other studies show that photolysis of ClNO2 during the daytime can lead to a significant source of radicals and NOx. There is no mention of ClNO2 throughout the manuscript, nor its potential impact on the diurnal profile. I recommend looking closely at these reactions in the model. Sufficient experimental data exists by now to parameterize N2O5 + Cl- chemistry on sea salt aerosol in the model.

Lastly, I feel the authors should clarify what parameters they are using to derive the nitrate photolysis rates. Are the absorption cross sections and quantum yields for gas phase nitric acid or aqueous nitrate used? I do not think it would be correct to use gas phase nitric acid parameters to derive photolysis rate constants when the focus is on aqueous (particulate) nitrate as the daytime renoxification source. After all, HNO3 is a strong acid and will be present as nitrate on aerosol surfaces or in bulk aqueous droplets under atmospherically relevant conditions found in the field. Aqueous nitrate photochemical parameters are therefore most accurate and applicable to this study.

Specific Comments

Abstract and P2: L10 – I suggest defining the acronym "PAN" when it is first mentioned in the abstract and in the main manuscript.

P2: L17 – Remove the word "through"

P3: L10 – Remove the first "global"

P3: L23-25 – Request for clarification: If the calibration is done in ambient air (rather than in zero air) how can one be sure what the exact concentration is. Are standard additions of NO and NO2 done for calibration?

P4: L21 – Do the authors mean: "so as not to sample from the main lab manifold".

P9: L2: delete "is." L13: the authors state, the major net sink for NOx is the formation of nitric acid by reaction of NO2 and OH. What about N2O5 deposition to aerosols as a major source of HNO3?

P12: L3 – add "cycle" or "profile" after diurnal.

Figure 2. Shaded area indicating standard deviation of the measurements does not show up on my copy. Consider using a different color (e.g., black and grey).

Figure 6: HNO3 photolysis is listed as a source of NO3 or OH and NO2. Is this formation rate considering a 10 fold enhancement of the HNO3 (or aq. nitrate) photolysis rate, or is this just un-scaled HNO3 photolysis using quantum yields and x-sections from JPL evaluations?
* * *

---

## Author Response (AR1)

1 Author's response to reviewer 1 2

3 The authors would like to thank the reviewer for their careful review and positive comments on the significance and 4 robustness of this manuscript and for taking the time to review our work. Our responses to their points are detailed 5 point-by-point below.

- 6 7 General comments:
- 8

9 The Authors present a compelling set of model results to explain the chemistry underpinning commonly observed 10 daytime maxima in NOx at the Cape Verde Atmospheric Observatory. The impact of condensed phase nitrate 11 photolysis in improving understanding of the NOx cycle in the marine boundary layer from long-term datasets, which capture diurnal features, has not been presented to date. This manuscript provides a robust method with which 12 13 to test the findings of intensive field and lab observations of this phenomenon. The authors also find that the role of 14 halogens is important in describing several features in the temporal nature of the NOx diurnal patterns, building 15 on recent findings that such chemistry may be important in controlling the cycling of reactive nitrogen in remote 16 marine regions. The presented manuscript is well-written and most of the data in the figures is presented clearly. 17 Overall, this work is acceptable for publication in Atmospheric Chemistry and Physics after a number of minor revisions and technical corrections have been made, which are presented in detail below.

18 19

20 Minor comments: 21

22 1) The title of the manuscript does not convey two of the major topics of this paper, nitrous acid and halogen 23 hydroxides. The authors should consider modifying their title to reflect the important roles of HONO and halogens 24 on renoxification processes in this work.

25

26 Incorporating all the major aspects of the paper would make for a very long title. We prefer to leave it as it – the 27 information on key aspects is in the abstract.

28

29 2) The authors cite a number of real-world surface (Baergen and Donaldson, 2013; Ye et al., 2016a), laboratory 30 substrate studies (Handley et al., 2007; Scharko et al., 2014; Zhou et al., 2003), a model estimate (Cohan et al., 31 2008) and two works on aerosol nitrate catalytic degradation (Ndour et al., 2009) and photolysis (Ye et al., 2016b) 32 as the basis for parameterizing the particulate nitrate conversion rates in their model (e.g Pages 7-8). The majority of 33 the cited work is for nitrate photolysis on proxy surfaces at the atmospheric interface and this is not clearly stated 34 throughout the manuscript, which makes the focus on aerosol nitrate photolysis throughout the manuscript 35 somewhat confusing. The connection and implications of linking nitrate photolysis on/in these other condensed 36 phases is not clear in its applicability or in its limitations and this would be worth expanding on in the manuscript. 37 This photochemistry is obviously important in this environment, but if studies of surface media and bulk aqueous 38 solution (Scharko et al., 2014) are used to constrain the rates in the model and contribute to the discussion (e.g. 39 effects of pH and RH), then the discussion should be expanded to include the expected role of surfaces versus 40 aerosols in the MBL or how the parameterization of aerosol photolysis encompasses all of these sources. 41

42

We agree, and have expanded this discussion as follows: 43

44 Changed (pg 9): "There have been a number of papers which have identified much faster photolysis of nitrate within 45 and on aerosol, than for gas phase nitric acid (Baergen and Donaldson, 2013; Cohan et al., 2008; Handley et al., 46 2007; Ndour et al., 2009; Scharko et al., 2014; Ye et al., 2016a, 2016b; Zhou et al., 2003).

47

48 To: "There have been a number of studies that have identified much faster photolysis of nitrate within and on

49 aerosol, than for gas phase nitric acid. These include studies using real-world natural and artificial surfaces

50 (Baergen and Donaldson, 2013; Ye et al., 2016a), laboratory substrates such as organic films and aqueous acidic

51 solutions (Handley et al., 2007; Scharko et al., 2014; Zhou et al., 2003), aerosol nitrate (Ndour et al., 2009; Ye et al.,

52 2016b), and a model estimate (Cohan et al., 2008). "

"The product ratio appears dependent on aerosol pH (Scharko et al., 2014)" to "The product ratio appears dependent
 on aerosol pH, with HONO production occurring only at low pH (Scharko et al., 2014).""

3 4 5

6

7

After "In order to explore the implications for Cape Verde NOx chemistry, we re-ran the base model removing the PAN source but including particulate nitrate (p-NO3) photolysis (R6) leading to HONO and NO2 production, scaled to the gas phase photolysis of HNO3." We have added "This parameterisation nominally represents photolysis of nitrate within and on aerosol, however conceptually includes any additional surface production of HONO and NO2."

3) The rapid photolysis of aerosol nitrate suggests that the lifetime of the reservoir may be quite short during the day
(~hours, (Ye et al., 2016b)), but this may be dependent on the chosen photolysis rate. It would be worthwhile to
discuss this and present the diurnal trend from the model (or indicate that this term is held constant) to compare to
these previous findings. Clarifying whether the aerosol nitrate photolysis mechanism can operate on the ambient
aerosol observed without depleting it and discussing how the reservoir is maintained would improve the argument

14 that this is a reasonable HONO and NOx source.

15

16 All model parameters are unconstrained, that is they are initialised at the stated values and allowed to reach

equilibrium which occurs within 3 days of starting the model (with a 1 second step size). This is stated in the model

18 description. Because nitrate is in large excess to the  $NO_x$  formed, our model simulations show no significant

depletion of the aerosol nitrate. We have added particulate nitrate to figure 10 (of model simulations of the diurnal

20 behaviour of NOy) to demonstrate the conservation of particulate nitrate through the model simulations.

21

4) The (Crilley et al., 2016) manuscript only cites a (Heland et al., 2001) paper on the LOPAP, without any

23 operational details on how such low detection limits were achieved for the instrument used in this work. The

24 majority of the data in Figure 2 are below the stated LOD of the LOPAP (< 1pptv), suggesting that this data has an

associated high uncertainty, which is not depicted. What is the exact LOD of the instrument and what data can be reliably reported in this figure? It would also be helpful to present the methods for calibration, background

correction, and determining the precision and accuracy of the measurements, as those achieved here are non-trivial.

We have updated Section 2.2 to include more details on the operation of the LOPAP at Cape Verde. At CVO, the
sampling conditions were set in order to maximise the sensitivity of the LOPAP, using a gas sampling flow rate of 2

31 lpm. A 2 point calibration was performed using a standard solution of nitrite (NO2-) at concentrations of 0.8 and 10

32 µg L-1. To account for instrument drift, baseline measurements using an overflow of high-purity N2 were performed

at regular intervals (8 hours). The detection limit  $(2\sigma)$  of the LOPAP was calculated by the variability during a

typical baseline measurement under N2 and was found to be 0.2 pptV. The relative error of the LOPAP was conservatively set to 10% of the measured concentration.

36

We have included the details of how we performed the calibration, baseline corrections and calculation of thedetection limit at Cape Verde, so that Section 2.2 now reads:

39

40 "Between 24th November and 3rd December 2015 a Long Path Absorption Photometer (LOPAP) (Heland et al.,

41 2001) was employed at CVO to provide an in-situ¬ measurement of nitrous acid. The LOPAP has its own

42 thermostated inlet system with reactive HONO stripping to minimise losses so did not sample from the main lab

43 manifold. The LOPAP inlet was installed on the roof of a container lab  $\sim 2.5$  m above ground level, unobstructed

from the prevailing wind. Calibration and operation of the LOPAP was carried out in line with the standard

45 procedures described by Kleffmann and Wiesen, (2008). Specifically at CVO, the sampling conditions were set in 46 order to maximise the sensitivity of the LOPAP, using a gas sampling flow rate of 2 lpm. A two point calibration

order to maximise the sensitivity of the LOPAP, using a gas sampling flow rate of 2 lpm. A two point calibration
 was performed using a standard solution of nitrite (NO2-) at concentrations of 0.8 and 10 μg L-1. To account for

47 was performed using a standard solution of mutie (NO2) at concentrations of 0.8 and 10 µg L-1. To account for 48 instrument drift, baseline measurements using an overflow of high-purity N2 were performed at regular intervals (8)

hours). The detection limit of the LOPAP ( $2\sigma$ ) was calculated by the variability during a typical baseline

50 measurement under N2 and was found to be 0.2 pptV. The relative error of the LOPAP was conservatively set to 51 10% of the measured concentration."

52

53 In addition, with regards to Figure 2, in reviewing the data we noticed an error in the baseline corrections applied, 54 with the updated figure shown below. From the new Figure 2, the majority of the data is now above the detection 1 limit (0.2 pptV) for the LOPAP, and so will have the associated uncertainty previously stated (10%). The net effect 2 is a small, but appreciable improvement in model/observation comparison.

3

5

7

8

9

11

5) The Authors focus their model on the 'summer season' (Page 5, Line 3) as this is the period of greatest data 4 density from CVO. Is this dataset also filtered for clear-sky days to reduce comparison bias between the model and 6 measurements? This season is most likely to be affected by cloudy days according to CVO observations reported in (Carpenter et al., 2010). Also, the model compares to HONO measurements from the winter period, when ambient NOx measurements do not exhibit the same diurnal pattern (i.e. the mid-day maximum) that the majority of this manuscript seeks to explain. It would be useful to present why the authors expect that winter HONO mixing ratios 10 and diurnal structure are representative of summer HONO.

12 Data has not been filtered for cloud cover due to the rapid nature of the chemistry involved and the low time

13 resolution (twice per day) cloud cover data provided by the Mindelo weather station which is ~15 km away over

14 hilly terrain. The average cloud cover for the summer period was 45%, consisting of predominantly broken cumulus 15 clouds moving at speed.

16 We focus on the summer season as it has the greatest data coverage and is out of the dust season which runs through

winter and spring (Carpenter et al., 2010; Fomba et al., 2014). The period of HONO measurements occurs in a dust 17

18 free period, while the majority of the winter  $NO_x$  measurements are heavily influenced by dust which has a greater

19 effect on photolysis rates and cloud cover. The more or less constant nitrate concentrations over the entire year and

20 the relatively small seasonal changes in solar radiation at this tropical location (and in fact temperature, wind speed

21 and direction etc) (Carpenter et al., 2010) lead us to believe that it is reasonable to expect HONO abundances and

22 behaviour to be similar in winter and summer. A midday maximum in  $NO_x$  is observed across all seasons at the

23 CVO (although some data are noisier), so we have evidence that the process is occurring year-round.

24 We have added greater explanation and detail to this effect. 25

26 6) Finally, how have the authors included or reasonably excluded boundary layer and transport dynamics in their

27 0D model? The report from (Carpenter et al., 2010) states that the limited information available in this regard

28 indicates no diel modulation of the boundary layer height, but that it can change substantially from day to day at a

29 site 200 km away. The work cited for DSMACC (Emmerson and Evans, 2009) does not suggest how the boundary

30 layer is represented in the model and it could be that some of the mismatch in early morning and evening NOx levels

31 is due to mixing and entrainment or local transport phenomena instead of the model chemistry. It would be useful

32 for the authors to discuss such processes as being accounted for, or as limitations, in the methods and at the

33 appropriate points in the manuscript discussion (e.g. (Wolfe et al., 2016)).

34

The boundary layer is fixed in the DSMACC model at the average cloud base height as reported in Carpenter et al., 35

(2010) which is expected to approximate boundary layer height. This is a reasonable approximation at a site 36

37 receiving maritime air as the sea surface temperature doesn't change much over the course of a day due to the large

- 38 thermal mass. This is in contrast to the study cited (Wolfe et al., 2016) by the reviewer which concentrates on
- 39 measurements over land with large daily variability which does indeed result a mismatch between model and 40 observation due to averaging.
- 41 It is conceivable that very rapid mixing between a layer with halogens and a layer without halogens could result in

42 the mismatch between model and observed  $NO_x$ , however, a mechanism to remove the halogens as quickly as 43 mixing occurred would also be needed.

- 44 We agree with the reviewer that it would be useful to include this discussion – and have added the following to
- 45 Section 2.3 describing the box model.
- 46 "The meteorological parameters pressure, temperature, relative humidity, and boundary layer height are set to
- 47 median values reported by Carpenter et al., (2010). Boundary layer height is fixed at 713m as no overall seasonal or
- 48 diel pattern is evident in boundary layer height at Cape Verde (Carpenter et al., 2010). This is entirely expected at a
- 49 site representative of the marine boundary layer, which has almost no island effects (except for very rare instances of

50 wind outside the northwesterly sector, which are excluded). Thus – we discount any influence from boundary layer

- 51 height changes on the diurnal cycles presented"
- 52 53 Technical corrections:

Page 1, Line 26: 'the box model simulation' of what? Everything?

"of NOx" added

1

2 3

4 5

6 7

8

14

Page 2, Line 3: provide a range of typical values for NOx observations in the remote MBL

Range added (10 to <100 pptV) with references. (Carsey et al., 1997; Lee et al., 2009; Monks et al., 1998)

Page 2, Lines 13-16: This is an example where the authors specify only particulate nitrate, yet cite work probing a variety of condensed phase nitrate proxies, ranging from surface-adsorbed nitrate to bulk aqueous solutions. The authors should be more specific here regarding the media and interfaces (e.g. particle-gas, surface-gas, aqueous gas) these works have described and that they have all found an enhancement in nitrate photolysis in the condensed phase although the mechanisms are not well understood.

Agreed, as with point 2 above we have clarified this section to be more specific about which surface/phase eachstudy refers to. Changed to:

''However, more recently the possibility of 'renoxification' by rapid nitrate photolysis on a variety of surfaces has
garnered attention. Photolytic rate enhancements have been reported on aerosol nitrate (Ndour et al., 2009; Ye et al.,
2016b), urban grime (Baergen and Donaldson, 2013, 2016), natural and artificial surfaces (Ye et al., 2016a), and in
laboratory prepared organic films and aqueous solutions (Handley et al., 2007; Scharko et al., 2014; Zhou et al.,
2003)."

Page 2, Line 17: Delete 'through'

26 Done 27

Page 2, Lines 17-21: Specify that these reactions are all taking place in the gas phase.

29 30 **Done**

28

31 32

33

34

Page 3, Line 17: 'for a short period in Nov/Dec 2015' should be restated to the number of days in the winter of 2015, with the observational period explicitly given in the HONO measurement section.

Done – the observational period was already explicitly given in the first line of the measurement section.
"for 10 days in Winter 2015" added

38 Page 3, Line 23 - Page 4, Line 4: Is assessment of RH and O3 effects on NO sensitivity and NO2 converter 39 efficiency at a period of 71 hours assuming that there is little change in sample RH and O3 over time or that the 1 40 hour offset in performing this calibration, spread over 2 years, corrects for these diurnal variations over the long term? Also, the measurements reported by (Lee et al., 2009) indicate that this period was 37 hours long. The authors 41 42 describe in detail how RH of the sample flow is minimized, but do not present information as to the range or relative 43 proportion of RH that sample flows are reduced to/by. O3 has a clear diurnal cycle presented throughout the manuscript, so it would be expected that corrections are necessary on an hourly timescale, not once every three days. 44 45 While many other interferences are clearly detailed for the approach to correction in (Lee et al., 2009; Reed et al., 46 2016a, 2016b) this particular modification and approach could be more clearly demonstrated to have little variation, 47 with an example of the variability that would be required to have significant systematic bias in the measurement due 48 to RH and O3 changes within a 71 hour period. Also, the authors do not present any information about whether 49 aerosols are removed from the sample flow, which could lead to artifact NOx signals in the system, similarly to 50 other adsorbed species in the photolysis cell. A greater description of the main lab manifold at the beginning of this 51 section would be sufficient to clarify. 52

A description of the lab manifold has been added, as well as including details of the sample filtration (0.22 micron
 filter) used.

- "Air is sampled from a common 40 mm glass manifold (QVF) which draws ambient air from a height of 10m above ground level. The manifold is downward facing into the prevailing wind at the inlet and fitted with a hood. The
- 3 ground level. The manifold is downward facing into the prevailing wind at the inlet and fitted with a hood. The 4 manifold is shielded from sunlight outside, and thermostated within the lab to  $30^{\circ}$ C to prevent condensation. Air is
- 5 drawn down by centrifugal pump at ~ 750 L/min-1 resulting in a sample flow speed of 10 m/s-1 and a residence time
- 6 to the NOx instrument of 2.3 seconds. Humidity and aerosol are reduced by two dead-end traps at the lowest points
- 7 of the manifold inside and outside the lab which are drained off regularly. Manifold sample flow, humidity and
- 8 temperature are recorded and logged continuously.
- 9 Air is sampled a  $90^{\circ}$  to the manifold flow through 1/4 inch PFA tubing at 1 standard L per minute, being filtered
- 10 through a 47mm, 0.22 μm mesh filter before entering the NOx analyser."
- 11

Regarding changing  $O_3$  biasing the converter efficiency, the high photolysis power converter reduces conversion efficiency by 0.013% per ppb  $O_3$ . The seasonal range in this study is ~ 11ppb ozone resulting in a 0.14% variation over the year, whereas the maximum daily variation in  $O_3$  reported by (Read et al., 2008) is 5 ppb, so 0.065%

- 15 change in NO2 conversion efficiency due to ozone change. This is well within the accuracy of the overall
- 16 measurement uncertainty.
- 17

18 Regarding sample drying and variability, the Rh% at Cape Verde can vary between ~60 to 90% (Carpenter et al.,

- 19 2010) which would have a dramatic effect on sensitivity through quenching of the chemiluminescent reaction and
- only in so much that sensitivity drift is between calibrations is <2% between maintenance periods. This point has been added after the description of the Nafion dryer.
- 22 23

24 "The humidity of the sample gas is further reduced by a Nafion dryer (PD-50T-12-MKR, Permapure), fed by a 25 constant sheath flow of zero air (PAG 003, Eco Physics AG) which is also filtered through a Sofnofil (Molecular 26 Products) and activated carbon (Sigma Aldrich) trap. This reduces sample humidity variability which affected NO 27 sensitivity through chemiluminescent quenching (Clough and Thrush, 1967) where sample humidity can vary from 28 60 to 90% (Carpenter et al., 2010). Calibration for NO sensitivity and NO2 converter efficiency occurs every 73 29 hours in ambient air as described by Lee et al., (2009); in this way correction for humidity affecting sensitivity, and 30 O3 affecting NO2 conversion efficiency are unnecessary. Sensitivity drift between calibration is < 2%, within the 31 overall uncertainty of the measurement."

32

A correction; NOx calibration was every 73 hours, rather than 71 as originally stated. Regarding the change from 37 hours (Lee et al., 2009), in that work the sample was not dried and the instrument sampled from an external inlet housing the NO2 and an NOy converter which were subject to heating during the day. For these reasons calibration was required more frequently. Later upgrades improved the stability greatly through better temperature control and gas handling requiring less frequent (lengthy) calibration.

38

Page 4, Line 15: Should 'period' be 'range'? 'being' should be 'are' and since much of this section is reporting data
 to two significant digits, shouldn't the detection limit for NO of 0.3 be 0.30?

4142 Period is correct, LOD of NO has been corrected to 0.30.

43
44 Page 4, Line 21: The 'main lab manifold' is not described above. It would be very useful to have this presented
45 above to know how external air is being delivered to the NOx instrumentation.

- A description of the lab manifold, its flow rate, diameter, material etc has been added to the description of NO and
   NO2 measurements as per a previous point.
- 49
- 50 Page 4, Lines 25-26: Do the PM measurements at the site ever indicate the presence of nitrite? Given the prevalence
- of dust impacting the site, nitrite could be formed on these surfaces. The LOPAP has been shown to effectively sample large aerosols, such as fog droplets, and the authors state dust and sea salt as dominating the mass transport

46

- 1 of condensed nitrate to CVO. This could bias the HONO measurement as LOPAP instrumentation does not typically 2 exclude such coarse particles (e.g. (Sörgel et al., 2011)) and the dual-channel scrubbing coils used to quantify 3 background and interference signals only effectively transmit particles less than 1 micrometer in diameter. 4 5 Neither the long term study of Fomba et al., (2014) nor the short term study of Muller et al., (2010) at the Cape 6 Verde site report the presence of nitrite in aerosol. 7 8 Page 6, Line 6: should this sentence finish 'in the instrument inlet'? From (Reed et al., 2016a, 2016b) the thermal 9 decomposition of PAN seems to occur in the photolysis cell? 10 11 No, the Lee et al., (2009) paper attributes the level of NOx to NOy species decomposing in the atmosphere. The word 12 "atmospheric" has been added to clarify this. 13 14 Page 6, Lines 10-11: This sentence is describing nocturnal processes, yet photolysis and OH losses are listed. Please 15 correct this error. Also, there is evidence in the presented data that the HONO buildup at night is still occurring (data 16 below 0 pptV at 18:00, and above at 06:00) as would be expected, from the measured precursor NO2 being present 17 at night and able to undergo heterogeneous hydrolysis. This may not be statistically significant, depending on the 18 uncertainty in the HONO measurement, or the data may only be an estimate based on the exact instrument detection 19 limits, so some clarification here should be given by considering those two limits. Previous work has also shown a 20 rapid approach to steady state in nocturnal HONO in marine environments due to reversible thermodynamic 21 partitioning in marine boundary layer surface waters, which is not mentioned here (Wojtal et al., 2011). 22 23 In response to the reviewers 4th point we have specified the measurement uncertainty and LOD for HONO. We have 24 added discussion of the nocturnal steady state concentration of HONO with reference to the reviewers suggested 25 reference. This paragraph now reads: 26 27 "Figure 2 shows the average diurnal cycle at CVO of measured HONO concentrations. The data exhibits a strong 28 daytime maximum peaking at noon local time (Solar noon ~13:20) and reaching near zero at night, indicating a 29 photolytic source. HONO is lost through deposition, photolysis and reaction with OH, whilst night time build-up often observed (Ren et al., 2010; VandenBoer et al., 2014; Zhou et al., 2002), here HONO appears to reach a steady 30 31 state concentration of ~0.65 pptV throughout the night. This pseudo steady state behaviour of nocturnal HONO has 32 previously been reported in the polluted marine boundary layer by Wojtal et al., (2011), albeit reporting much higher 33 HONO mixing ratios." 34 35 Page 6, Line 17: 'daytime' should be placed between 'additional' and 'source'
- 3637 Added
- 39 Page 6, Lines 19-20: 'are difficult to explain' should be 'cannot be explained'
- 40 41 Corrected
  - Page 6, Line 21: 'either of NOx or HONO'. Shouldn't this be 'NOx and HONO'?
- 4445 Corrected
- 46

- Page 7, Lines 22-23: The authors should replace 'would appear' with 'is'. Also, it would seem that the intrusion of
  ship emissions, if stochastic, would be normalized from the mean through the consideration of 2 years of summer
  data. This is supported by the range versus the mean of the NOx data presented in many figures.
- 50 51 Agreed, change made.
- 52

- Page 8, Lines 14-17: It would be expected that the aerosol nitrate would be distributed across both fine and coarse
  mode aerosol and photolyze differently based on their optical and chemical properties. The authors state in lines 2628 that this is the case. It would be useful to clarify that the best match of nitrate photolysis enhancement that
  reproduces observed HONO is a rate integrated across all surface nitrate photolysis sources at CVO since only bulk
- aerosol composition has been measured in (Fomba et al., 2014).
- Agreed. We have added in the paragraph immediately below on page 9 lines 4,5 that we parameterize all aerosol nitrate and aerosol surface area. The uptake of HNO3 or XONO2 aerosol surface forms 'NIT' + some other species which is then photolysable at a single faster rate which is a multiple of the gas phase HNO3 *j*value.
- "This parameterisation nominally represents photolysis of nitrate within and on aerosol, however conceptually
   includes any additional surface production of HONO and NO2."
- 13

47

14 Page 8, Lines 26-28: It is confusing why the authors cite the (Laufs and Kleffmann, 2016) work here as they state in 15 the abstract of their work, a conclusion counter to the thesis of this work: 'If these results can be translated to 16 atmospheric surfaces, HNO3 photolysis cannot explain the significant HONO levels in the daytime atmosphere. In 17 addition, it is demonstrated that even the small measured yields of HONO did not result from the direct photolysis of 18 HNO3 but rather from the consecutive heterogeneous conversion of the primary photolysis product NO2 on the 19 humid surfaces. The secondary NO2 conversion was not photoenhanced on pure quartz glass surfaces in good 20 agreement with former studies. A photolysis frequency for the primary reaction product NO2 of J(HNO3 - NO2) =21 1.1x10-6 s-1 has been calculated (0 SZA, 50% r.h.), which indicates that renoxification by photolysis of adsorbed 22 HNO3 on non-reactive surfaces is also a minor process in the atmosphere.' The work described by the cited works 23 of (Baergen and Donaldson, 2013, 2016; Scharko et al., 2014; Ye et al., 2016a, 2016b; Zhou et al., 2003) are all in 24 disagreement with (Laufs and Kleffmann, 2016) and the photolysis rates from these measurements are used to 25 constrain this model. They also clearly discuss the wide range of photolysis values without such contradictory

26 statements. The authors should consider revising the works cited in this location.
27

Agreed. Laufs and Kleffmann, (2016) was cited as a low end estimate of HNO3 photolysis frequency on surfaces,
 rather than for its overarching conclusion. The reference has been removed.

Page 9, Line 1: Figure 5 includes PAN transport. Remove the reference to it here.

Page 9, Lines 5-8: The ability to reproduce the NOx profile is based on a large loss of NO2 and production of NO,
 the former of which is not observationally consistent. Stating this and the need to explore further chemical
 mechanistic constraints would improve the transition to the next section of the manuscript.

37 Agreed, the following paragraph has been added immediately before section 3.3

38 39 "Introduction of an additional source of  $NO_x$  is able to roughly produce a flat diurnal cycle, though is not able to 40 simulate a definite peak of  $NO_x$  during daytime. With the addition of a source and no change in sinks for NOx this is 41 unsurprising and leads to over estimation of  $NO_x$ . This is therefore likely that one or more NOx sinks are absent 42 from the base simulation which must be explored further."

Page 9, Line 9: It would be useful to include some reference to halogen chemistry in this section header 45

- 46 A short introduction to halogen nitrate formation has been added.
- 48 "Aside from loss to  $HNO_3$  directly through reaction with OH (R1)  $NO_x$  is also lost to nitrate by reaction with
- 49 halogen oxides (XO) forming a halogen nitrates (R7) (Keene et al., 2009). Read et al., (2008) showed how halogen

50 oxides mediate ozone formation and loss at Cape Verde thus their indirect effect on  $NO_x$ . Their direct effect on  $NO_x$

- 51 loss was included in studying  $NO_x$  sinks.
- $52 \quad XO + NO_2 + M \rightarrow XONO_2 + M \quad (R7)"$
- 54 Page 9, Line 12: '(NIT)' this is the only instance of this shorthand in the manuscript.

| 1                          | Delete.                                                                                                                                                                                                                                                                                                                                                                                                   |
|----------------------------|-----------------------------------------------------------------------------------------------------------------------------------------------------------------------------------------------------------------------------------------------------------------------------------------------------------------------------------------------------------------------------------------------------------|
| 2
3                     | Done                                                                                                                                                                                                                                                                                                                                                                                                      |
| 4
5
6                | Page 9, Lines 13-15: This would be much easier to follow if broken into 2-3 sentences.                                                                                                                                                                                                                                                                                                                    |
| 7                          | Agreed, reworded for clarity                                                                                                                                                                                                                                                                                                                                                                              |
| 9
10
11              | Page 10, Lines 5-6: Dust and sea salt are stated to be the 'predominant aerosol' at CVO. Is this by number, mass, or surface area? Please specify, with reference to (Carpenter et al., 2010; Fomba et al., 2014), so there is greater clarity in understanding if the majority of the nitrate is expected to be in the coarse mode.                                                                      |
| 12
13
14             | By mass, in coarse mode added with reference to Fomba et al., (2014) and Carpenter et al., (2010)                                                                                                                                                                                                                                                                                                         |
| 15
16
17             | Page 10, Line 14: Delete 'e.g. JPL' and change the citation format to 'Burkholder et al., (2015)'                                                                                                                                                                                                                                                                                                         |
| 18
19                   | Done                                                                                                                                                                                                                                                                                                                                                                                                      |
| 20
21
22             | Page 10, Lines 15-16: It would seem that the heterogeneous chemistry on fine mode aerosol may be what is poorly constrained. Would it be possible to speculate on this?                                                                                                                                                                                                                                   |
| 23
24
25
26       | Indeed, as noted by Abbatt et al., (2012) uptake coefficients of many reactive uptake processes are very poorly constrained, in addition to gaps in our understanding of gas phase halogen chemistry highlighted by Simpson et al., (2015).                                                                                                                                                               |
| 20
27
28
29
30 | Page 10, Line 19: 'NO3' should be 'HNO3'. Also, is the static reactive uptake coefficient of 0.15 used in the model for HNO3 partitioning reasonable given the likely need for this value to increase mid-day to sustain the reservoir of particulate nitrate?                                                                                                                                            |
| 31
32
33
34       | Corrected to $HNO_3$ . A static uptake coefficient is a reasonable assumption (without information to the contrary) in this case as nitrate is minimally depleted compared to the total amount during daytime as shown in response to a previous comment.                                                                                                                                                 |
| 35
36                   | Page 12, Lines 8-9: This seems like a transition to an 'Atmospheric Implications' section                                                                                                                                                                                                                                                                                                                 |
| 37
38                   | Page 12, Line 18: Update this to include the role of other surfaces.                                                                                                                                                                                                                                                                                                                                      |
| 39
40                   | This paragraph has been reworded to be less specific about aerosol nitrate and included other possible surface sources.                                                                                                                                                                                                                                                                                   |
| 41
42
43
44
45 | "From these simulations it would appear that the photolysis of surface adsorbed nitrate may be the dominant source
of NOx into the marine boundary layer around Cape Verde. Photolysis of aerosol nitrate, or nitrate in solution would
be capable of producing a diurnal cycle in NOx which was consistent with the observations when HOX + NO3
chemistry is considered also."                  |
| 46
47
48
49       | References: Chemical subscripts and capitalization issues need to be corrected in: Burkholder et al (2015), Evans and Jacob (2005), Handley et al (2007), Laufs and Kleffmann (2016), Li et al (2014), Moxim et al (1996), Nakamura et al (2003), Pollack et al (2011), Ryerson et al (2000), Saiz-Lopez et al (2008), Sander et al (1999), Scharko et al (2014), Ye et al (2016a), and Zhou et al (2003) |
| 50
51
52             | Corrected                                                                                                                                                                                                                                                                                                                                                                                                 |
| 53                         | Figure 1: Why is the NOx axis red, when the NOx trace is black? The color scheme here is generally not suitable for                                                                                                                                                                                                                                                                                       |

Figure 1: Why is the NOx axis red, when the NOx trace is black? The color scheme here is generally not suitable for red-green color blind individuals and also does not print well in grayscale. Consider a scheme for figures, to use

1 throughout, that is more easily discerned. Standard error is weighted by the number of samples considered, but those 2 values are not presented anywhere. It would be worthwhile to do so, especially for the summer period. The rest of 3 the manuscript only considers the summer observations. Thus, only 'summer' requires a definition of the months 4 considered. Labels in the figure could just be the months considered and would remove the need to cross-reference. 5 6 The number or samples for the summer period was 153 for each hourly average data point. 7 We have changed the colour of the  $NO_x$  axis in this and all other figures to black and changed the figure labels to 8 define the months in each season. 9 10 Figure 2: Add the cumulative accuracy and precision error and depict the instrument 11 detection limit. 12 13 In reviewing the data we noticed an error in the baseline corrections applied which we have now corrected. From the 14 new Figure 2, the majority of the data is now above the detection limit (0.2 pptV) for the LOPAP, and so will have 15 the associated uncertainty previously stated (10%). The LOD has been indicated also. 16 17 Figure 3: (left) For all plots like this, would it be more informative to present the values of the difference between 18 the model and the measurement? The color and formatting challenges noted in Figure 1 apply here too. (right) The 19 reaction text is difficult to read and the scale breaks are confusing. Would a log scale work and still emphasize the 20 necessary rates? 21 22 Agreed, we have added, rather that substituted a panel showing the difference between model and measurement for 23 NOx and HONO. Additionally, the ROPA panel has been changed to be friendlier to any colour-blind reader and the 24 reaction text has been emboldened. 25 26 Figure 4: This figure could be simplified if the difference between NOx, NO2, and NO relative to the observations 27 were depicted in three separate panels for the photolysis factors considered. It would also be a more quantitative 28 representation of which factor is most suitable. 29 30 Agreed, the figure has been simplified into a single panned showing the difference between model and observation 31 for  $NO_x$  for the six different photolysis rates. (NO and  $NO_2$  disagree by the same factor given the same oxidant 32 concentration). The original figure is moved to supplementary information. 33 34 Figure 6: Can the magnitude of the particulate nitrate photolysis be presented here? It would be nice to compare it to 35 the other NOx source mechanisms. Also, it is surprising that HONO photolysis isn't presented as the manuscript 36 suggests that its intermediate nature is key in reNOxification at CVO. (right) Same comments as Fig 3. (caption) 37 Insert 'for NOx' after 'loss analysis' 38 39 The figure has been updated with new colours and bolder text. With regards to the reviewers first point te magnitude 40 of nitrate photolysis (p-NO3  $\rightarrow$  NO2/HONO) is presented in the right panel and constitutes the top two major sources 41 of NOx. We have now made this more clear in the text. 42 43 Figure 7: There is no PAN on this figure, but it is listed in the caption. The difference notation, again, may be more 44 informative for presenting the comparisons. 45 46 Agreed, the difference notation has been used to show model-observation disagreement for NOx, HONO, IO and 47 BrO in place of the original figure which is moved to supplementary material. Reference to PAN has been removed. 48 49 Figure 8: Could the dips in the early morning NOx in the model be mismatching the observations because of NOx 50 transport or dilution that isn't accounted for in the 0D model? 51 As in answer to the reviewers 6th point regarding boundary layer height and mixing in the 0-D model used: The 52 boundary layer is fixed in the DSMACC model at the average cloud base height as reported in Carpenter et al., 53 54 (2010) which is expected to approximate boundary layer height. This is a reasonable approximation at a site 9

- receiving maritime air as the sea surface temperature doesn't change much over the course of a day due to the large
   thermal mass.
- 4 Figure 11: What do the dashed lines represent?
- 5 Dashed lines represent HOx and OH the where dominant source of NOx is particulate nitrate photolysis and HOX + NO3 chemistry is included. The caption has been corrected indicating this.
- 89 Abbatt, J. P. D., Lee, A. K. Y. and Thornton, J. A.: Quantifying trace gas uptake to tropospheric aerosol: recent
- 10 advances and remaining challenges, Chem. Soc. Rev., 41(19), 6555, doi:10.1039/c2cs35052a, 2012.
- 11 Carpenter, L. J., Fleming, Z. L., Read, K. A., Lee, J. D., Moller, S. J., Hopkins, J. R., Purvis, R. M., Lewis, A. C.,
- 12 Müller, K., Heinold, B., Herrmann, H., Fomba, K. W., Pinxteren, D., Müller, C., Tegen, I., Wiedensohler, A.,
- 13 Müller, T., Niedermeier, N., Achterberg, E. P., Patey, M. D., Kozlova, E. A., Heimann, M., Heard, D. E., Plane, J.
- M. C., Mahajan, A., Oetjen, H., Ingham, T., Stone, D., Whalley, L. K., Evans, M. J., Pilling, M. J., Leigh, R. J.,
- 15 Monks, P. S., Karunaharan, A., Vaughan, S., Arnold, S. R., Tschritter, J., Pöhler, D., Frieß, U., Holla, R., Mendes,
- 16 L. M., Lopez, H., Faria, B., Manning, A. J. and Wallace, D. W. R.: Seasonal characteristics of tropical marine
- boundary layer air measured at the Cape Verde Atmospheric Observatory, J. Atmos. Chem., 67(2–3), 87–140,
- 18 doi:10.1007/s10874-011-9206-1, 2010.
- 19 Carsey, T. P., Churchill, D. D., Farmer, M. L., Fischer, C. J., Pszenny, A. A., Ross, V. B., Saltzman, E. S., Springer-
- 20 Young, M., Bonsang, B., Boss, V. B., Saltzmann, E. S., Springer-Young, M. and Bonsang, B.: Nitrogen oxides and
- ozone production in the North Atlantic marine boundary layer, J. Geophys. Res. Atmos., 102(D9), 653–665,
   doi:10.1029/96JD03511, 1997.
- 23 Fomba, K. W., Müller, K., Van Pinxteren, D., Poulain, L., Van Pinxteren, M. and Herrmann, H.: Long-term
- 24 chemical characterization of tropical and marine aerosols at the Cape Verde Atmospheric Observatory (CVAO)
- 25 from 2007 to 2011, Atmos. Chem. Phys., 14(17), 8883–8904, doi:10.5194/acp-14-8883-2014, 2014.
- Laufs, S. and Kleffmann, J.: Investigations on HONO formation from photolysis of adsorbed HNO 3 on quartz glass
- 27 surfaces, Phys. Chem. Chem. Phys., 18(14), 9616–9625, doi:10.1039/C6CP00436A, 2016.
- 28 Lee, J. D., Moller, S. J., Read, K. A., Lewis, A. C., Mendes, L. and Carpenter, L. J.: Year-round measurements of
- nitrogen oxides and ozone in the tropical North Atlantic marine boundary layer, J. Geophys. Res., 114(D21),
   D21302, doi:10.1029/2009JD011878, 2009.
- 31 Monks, P. S., Carpenter, L. J., Penkett, S. A., Ayers, G. P., Gillett, R. W., Galbally, I. E. and Meyer, C. P.:
- Fundamental ozone photochemistry in the remote marine boundary layer: The SOAPEX experiment, measurement
- and theory, Atmos. Environ., 32(21), 3647–3664, doi:10.1016/S1352-2310(98)00084-3, 1998.
- 34 Muller, K., Lehmann, S., van Pinxteren, D., Gnauk, T., Niedermeier, N., Wiedensohler, A. and Herrmann, H.:
- Particle characterization at the Cape Verde atmospheric observatory during the 2007 RHaMBLe intensive, Atmos.
   Chem. Phys., 10(6), 2709–2721, doi:10.5194/acpd-9-22739-2009, 2010.
- 37 Read, K. A., Mahajan, A. S., Carpenter, L. J., Evans, M. J., Faria, B. V. E., Heard, D. E., Hopkins, J. R., Lee, J. D.,
- Moller, S. J., Lewis, A. C., Mendes, L., McQuaid, J. B., Oetjen, H., Saiz-Lopez, A., Pilling, M. J. and Plane, J. M.
- C.: Extensive halogen-mediated ozone destruction over the tropical Atlantic Ocean, Nature, 453(7199), 1232–1235,
   doi:10.1038/nature07035, 2008.
- 41 Simpson, W. R., Brown, S. S., Saiz-Lopez, A., Thornton, J. A. and Von Glasow, R.: Tropospheric Halogen
- 42 Chemistry: Sources, Cycling, and Impacts, Chem. Rev., 115(10), 4035–4062, doi:10.1021/cr5006638, 2015.
- 43 Wolfe, G. M., Marvin, M. R., Roberts, S. J., Travis, K. R. and Liao, J.: The framework for 0-D atmospheric
- 44 modeling (F0AM) v3.1, Geosci. Model Dev., 9(9), 3309–3319, doi:10.5194/gmd-9-3309-2016, 2016.
- 45
- 46 Author's response to reviewer 2
- 47
- The authors would like to thank the reviewer for taking the time to assess our manuscript. We have answered their queries and suggestions point by point below.
- 50
- 51 Overall Assessment
- 52

and O3 concentrations in the marine boundary layer from measurements at a coastal site in Cape 2 Verde Atmospheric Observatory (CVO). Of particular interest is a noon-time high in NOx 3 4 concentrations. A box model approach was used to model this diurnal profile, which is explained as arising from particulate nitrate photolysis and reactions of reactions of halogen nitrates 5 (products of nitrate radical and halogen hydroxide chemistry during the nighttime). The authors 6 argue that field observations could not be explained by dissociation of PAN that is transported to 7 the site from anthropogenic sources over long distances. The methodology associated with 8 measurements of NOx, HONO, O3 etc. are appropriately chosen and carefully executed. I think 9 10 there is a potentially interesting data set here and a nice opportunity to explore the role of nitrate aerosol photochemistry as a daytime NOx source. However, for completeness I would like to see 11 an analysis of the relative importance of ClNO2 as a daytime NOx source vs. the other potential 12 daytime NOx sources that were postulated. 13 14 The authors mention that the diurnal pattern in the CVO NOx concentrations was historically 15 attributed to thermal decomposition of NOy species (see p. 6, L6). By NOy, the authors refer 16

The manuscript by Reed et al. presents 2-years of results on a unique diurnal cycle of NO, NO2

mostly to PAN, but what about N2O5 heterogeneous chemistry? Consideration of N2O5 17 heterogeneous chemistry appears to be limited to hydrolysis (modelled using N2O5==>2 NO3-, 18 with an uptake coefficient of 0.02). This likely explains why in Fig. 10 the model shows non-19 existent N2O5 concentrations at this site over a 24 hour period. However, previous studies of 20 N2O5 in coastal regions show that steady-state concentrations of 20-100 ppt can exist, with 21 peaks during the nighttime. Those studies also demonstrate that aside from hydrolysis to form 22 particulate nitrate, a major fate for N2O5 is conversion to ClNO2 on sea salt aerosol and the 23 ocean surface. (e.g. PNAS, 2014, 111, 3943). Other studies show that photolysis of ClNO2 24 during the daytime can lead to a significant source of radicals and NOx. There is no mention of 25 26 CINO2 throughout the manuscript, nor its potential impact on the diurnal profile. I recommend looking closely at these reactions in the model. Sufficient experimental data exists by now to 27 parameterize N2O5 + Cl- chemistry on sea salt aerosol in the model. 28

29

1

>While we agree with the reviewer that in their cited example N2O5 chemistry forming ClNO2
is a source of radicals and NOx at the Scripps pier (Kim et al., 2014), at Cape Verde NO2 is two

to three orders of magnitude lower than in California thus the equilibrium concentration of N2O5

is negligible. Furthermore the study of Savarino et al., (2013) specifically on the isotopic

- composition of nitrate at Cape Verde found isotope ratios which were incompatible with high
- production rates of HNO3 from N2O5 hydrolysis, and concluded that  $N_2O_5$  and nitryl compound
- 36 (CINO2 BrNO2) levels in this region are very low. This is consistent with other studies

37 modelling the pristine MBL at Cape Verde i.e. Sommariva and Von Glasow, (2012).<

- 38
- 39 Discussion of this has been added to discussion of figure 10. This now reads.
- 40
- 41 "In all cases N2O5 (in black) is effectively zero at all times due to very low NOx mixing ratios
- 42 in this pristine environment and the relatively high ambient temperatures (24.5 oC) where the
- 43 N2O5 lifetime is ~ 3 s-1. This precludes N2O5 channels to NOx (and ultimately nitrate),
- 44 consistent with the experimental findings of Savarino et al., (2013) at Cape Verde who found
- 45 isotope ratios which were incompatible with high production rates of HNO3 from N2O5

hydrolysis, and concluded that N2O5 and nitryl compound (ClNO2, BrNO2) levels in this region 1 are very low. This is consistent with our own and other studies modelling the pristine marine 2 boundary layer at Cape Verde of Sommariva and Von Glasow, (2012). This is in contrast with 3 4 more polluted regions where N2O5 has been shown to be a route to NOx and CINO2 (Kim et al., 2014)."< 5 6 Lastly, I feel the authors should clarify what parameters they are using to derive the nitrate 7 8 photolysis rates. Are the absorption cross sections and quantum yields for gas phase nitric acid or 9 aqueous nitrate used? I do not think it would be correct to use gas phase nitric acid parameters to 10 derive photolysis rate constants when the focus is on aqueous (particulate) nitrate as the daytime renoxification source. After all, HNO3 is a strong acid and will be present as nitrate on aerosol 11 surfaces or in bulk aqueous droplets under atmospherically relevant conditions found in the field. 12 Aqueous nitrate photochemical parameters are therefore most accurate and applicable to this 13 study. 14 15 > As stated in the text surface nitrate photolysis rates are scaled to that of gas phase nitric acid, 16 which is consistent with the results and approach of Ye et al., (2016) who found a correlation 17 between the required in situ HONO source and the product of the bulk nitrate concentration and 18 the photolysis frequency of gaseous HNO3. All surface nitrate was parameterized similarly as 19 bulk 'NIT' in our model study. This parameterisation in fact represents a convolution of what 20 could be many different surfaces or phases with many different quantum yields of which there 21 are poor constraints, thus we use the well-defined rate of gaseous HNO3 photolysis as a proxy. 22 23 **Specific Comments** 24 25 26 Abstract and P2: L10 – I suggest defining the acronym "PAN" when it is first mentioned in the abstract and in the main manuscript. 27 28 29 >'Peroxy acetyl nitrate' added< 30 P2: L17 – Remove the word "through" 31 32 33 >Removed< 34 P3: L10 – Remove the first "global" 35 36 37 >Removed< 38 39 P3: L23-25 – Request for clarification: If the calibration is done in ambient air (rather than in zero air) how can one be sure what the exact concentration is. Are standard additions of NO and 40 NO2 done for calibration? 41 42 43 >Correct, standard addition to ambient air is done for calibration. Line now reads:

| 1        | "Calibration for NO sensitivity and NO2 converter efficiency occurs every 73 hours by standard       |  |
|----------|------------------------------------------------------------------------------------------------------|--|
| 2        | addition to ambient air as described by Lee et al., (2009); in this way correction for humidity      |  |
| 3        | affecting sensitivity, and O3 affecting NO2 conversion efficiency are unnecessary."<                 |  |
| 4        |                                                                                                      |  |
| 5        | P4: L21 – Do the authors mean: "so as not to sample from the main lab manifold".                     |  |
| 6        |                                                                                                      |  |
| 7        | P9: L2: delete "Is." L13: the authors state, the major net sink for NOX is the formation of nitric   |  |
| 8        | actu by reaction of NO2 and OH. what about N2O3 deposition to derosols as a major source of          |  |
| 9        |                                                                                                      |  |
| 10       | See our response to the reviewers first point regarding N2O5 hydrolysis as a major source of         |  |
| 12       | HNO 2 <                                                                                   |  |
| 12       |                                                                                                      |  |
| 14       | P12: L3 – add "cycle" or "profile" after diurnal.                                                    |  |
| 15       |                                                                                                      |  |
| 16       | >Added<                                                                                              |  |
| 17       |                                                                                                      |  |
| 18       | Figure 2. Shaded area indicating standard deviation of the measurements does not show up on          |  |
| 19       | my copy. Consider using a different color (e.g., black and grey).                                    |  |
| 20       |                                                                                                      |  |
| 21       | >This figure is now changed<                                                                         |  |
| 22       |                                                                                                      |  |
| 23       | Figure 6: HNO3 photolysis is listed as a source of NO3 or OH and NO2. Is this formation rate         |  |
| 24       | considering a 10 fold enhancement of the HNO3 (or aq. nitrate) photolysis rate, or is this just un-  |  |
| 25       | scaled HNO3 photolysis using quantum yields and x-sections from JPL evaluations?                     |  |
| 26       |                                                                                                      |  |
| 27       | >This refers to purely gas phase photolysis of HNO3 and is not scaled. It does use photolysis        |  |
| 28       | quantum yields and corrections from JPL<                                                             |  |
| 29
20 | Deferences                                                                                           |  |
| 30       | References                                                                                           |  |
| 32       | Kim M I Farmer D K and Bertram T H · A controlling role for the air-sea interface in the             |  |
| 33       | chemical processing of reactive nitrogen in the coastal marine boundary laver. Proc. Natl. Acad.     |  |
| 34       | Sci., 111(11), 1–6, doi:10.1073/pnas.1318694111, 2014.                                               |  |
| 35       | Savarino, J., Morin, S., Erbland, J., Grannec, F., Patey, M. D., Vicars, W., Alexander, B. and       |  |
| 36       | Achterberg, E. P.: Isotopic composition of atmospheric nitrate in a tropical marine boundary         |  |
| 37       | layer, Proc. Natl. Acad. Sci., 110(44), 17668–17673, doi:10.1073/pnas.1216639110, 2013.              |  |
| 38       | Sommariva, R. and Von Glasow, R.: Multiphase halogen chemistry in the tropical atlantic ocean,       |  |
| 39       | Environ. Sci. Technol., 46(19), 10429–10437, doi:10.1021/es300209f, 2012.                            |  |
| 40       | Ye, C., Zhou, X., Pu, D., Stutz, J., Festa, J., Spolaor, M., Tsai, C., Cantrell, C., Mauldin, R. L., |  |
| 41       | Campos, T., Weinheimer, A., Hornbrook, R. S., Apel, E. C., Guenther, A., Kaser, L., Yuan, B.,        |  |
| 42       | Karl, T., Haggerty, J., Hall, S., Ullmann, K., Smith, J. N., Ortega, J. and Knote, C.: Rapid         |  |
| 43       | cycling of reactive nitrogen in the marine boundary layer, Nature, 532(7600), 489–491,               |  |
| 44       | doi:10.1038/nature17195, 2016.                                                                       |  |

**2 Evidence for renoxification in the tropical marine boundary layer**

Chris Reed1†, Mathew J. Evans1,2, Leigh R. Crilley3, William J. Bloss3, Tomás Sherwen1,
Katie A. Read1,2, James D. Lee1,2 and Lucy J. Carpenter1

[revised manuscript text omitted]

Formatte Subscrip Formatte Formatte Formatte **Deleted:** Formatte Formatte Subscrip Formatte Formatte Formatte Formatte Subscrip Formatte Formatte Formatte Formatte **Deleted: Deleted: Deleted: Deleted: Deleted: Deleted:**

Formatte

 $(\mathbf{R8})$

Increasing the  $\gamma$  of XONO2 from 0.02 (the low end of recommended values) to 0.1 results in 1 small changes to both the NOx and XO diurnals. The loss of NOx at sunrise becomes more 2 pronounced whereas the XO diurnals become slightly more 'square' or 'top-hat' as per the 3 observations of Read et al., (2008). Increasing the  $\gamma$  to the upper extreme ( $\gamma = 0.8$ ) results in a 4 spike in BrO at sunrise, which consumes the majority of NO2 though formation of BrONO2. No 5 combination of uptake coefficients can completely reproduce the characteristic XO diurnals due 6 7 to poor constraints on heterogeneous halogen chemistry (Abbatt et al., 2012) in addition to gaps in understanding of gas phase halogen chemistry (Simpson et al., 2015). 8

9 The effect on the NOx diurnal of changing  $\gamma$  is clear in that greater uptake coefficients 10 recommended by Burkholder et al., (2015) result in objectively worse simulation of both the NOx 11 and XO diurnals. It is therefore likely that information is lacking from the XO – NOx chemistry 12 scheme as it is currently known.

**13 **3.4 HOI/HOBr - NOx chemistry**

Recently, IO recycling by reaction with NO3 has been proposed by Saiz-Lopez et al., (2016), who calculated that the reaction (R9) of HOI + NO3 producing IO and HNO3 has a low enough activation energy and fast enough rate constant to be atmospherically relevant in the troposphere.

17 | HOI + NO3  $\rightarrow$  IO + HNO3 :  $k = 2.7 \times 10^{-12} (300/T)^{2.66}$

This mechanism provides a route to nitric acid, and thus particulate nitrate at night, whilst also leading to nocturnal IO production leading to loss of NO2 by IONO2 formation.

Including this new reaction and re-running the model leads to a diurnal profile of IO much more representative of the observations. This however introduces a more pronounced loss of  $NO_x$  at sunrise and sunset, and also results in  $NO_x$  peaking during the day which fits better with the observations as shown in Fig. 8. HONO is still underestimated during daytime though nocturnal values agree well.

The inclusion of this HOI + NO3 reaction reproduces the general NOx and O3 diurnals more closely than without i.e. a daytime maximum in NOx. There are also effects on the halogen oxide Formatte

(R9)

behaviour. The simulated BrO has a flatter profile, which more closely matches the observations.
 However, modelled IO is now non-zero at night and the sunrise build-up and sunset decay still
 occurs more abruptly than the observations.

Although the NOx and O3 diurnals are reproduced more closely with this new chemistry, there is still disagreement with the observed NOx diurnal at sunrise and sunset especially indicating a missing reaction or reactions. To best approximate the observed diurnal behaviour an analogous HOBr + NO3 night time reaction (R10) was introduced with a rate 10 times that of HOI + NO3 as calculated by Saiz-Lopez et al., (2016b)

9 HOBr + NO3  $\rightarrow$  BrO + HNO3 :  $k = 2.7 \times 10^{-11} (300/T)^{2.66}$  (R10)

This results in an improved reproduction of the observed  $NO_x$  diurnal, Fig. 9. This is a purely speculative representation in order to reproduce the observed  $NO_x$  diurnal and highlights how some mechanistic knowledge of  $NO_x$ -halogen-aerosol systems is still missing.

With HOX + NO3 chemistry included in the model as in Fig. 9, significant loss of NOx at sunrise 13 and sunset is eliminated and agreement is improved over any previous simulation. Greater 14 HONO production is also simulated, with up to ~ 3.0 pptV predicted – in line with the 15 observations shown in Fig. 2. Halogen oxide modelled diurnal cycles remain broadly consistent 16 with observations. Diagnosis of the net production and loss terms for NOx reveal that nitrate 17 photolysis to HONO or NO2 contribute ~ 80% of all NOx with decomposition of PAN 18 19 contributing the remainder. Major net sinks of  $NO_x$  are shown to be reaction with halogen hydroxides and OH to form HNO3. Nitric acid is then taken up on surfaces and recycled to  $NO_x$ 20 through photolysis to NO2 and HONO. 21

The improvement can be better understood by diagnosing the modelled NOy distribution. In Fig. 10 the distribution of PAN, IONO2, BrONO2, N2O5, NO3 and particulate nitrate (p-NO3) 
[revised manuscript text omitted]

- 4
- 5